# AF2Complex predicts direct physical interactions in multimeric proteins with deep learning

Mu Gao [1✉], Davi Nakajima An [2], Jerry M. Parks [3] & Jeffrey Skolnick [1✉]

Accurate descriptions of protein-protein interactions are essential for understanding biological systems. Remarkably accurate atomic structures have been recently computed for individual proteins by AlphaFold2 (AF2). Here, we demonstrate that the same neural network models from AF2 developed for single protein sequences can be adapted to predict the structures of multimeric protein complexes without retraining. In contrast to common approaches, our method, AF2Complex, does not require paired multiple sequence alignments. It achieves higher accuracy than some complex protein-protein docking strategies and provides a significant improvement over AF-Multimer, a development of AlphaFold for multimeric proteins. Moreover, we introduce metrics for predicting direct protein-protein interactions between arbitrary protein pairs and validate AF2Complex on some challenging benchmark sets and the *E. coli* proteome. Lastly, using the cytochrome *c* biogenesis system I as an example, we present high-confidence models of three sought-after assemblies formed by eight members of this system.

[1] Center for the Study of Systems Biology, School of Biological Sciences, Atlanta, GA, USA. [2] School of Computer Science, Georgia Institute of Technology, Atlanta, GA, USA. [3] Biosciences Division, Oak Ridge National Laboratory, Oak Ridge, TN, USA. ✉email: mu.gao@gatech.edu; skolnick@gatech.edu

AlphaFold2 (AF2), a deep learning approach developed by DeepMind for predicting protein structure given a sequence, has greatly advanced protein structure prediction[1,2]. In CASP14, a blind test for protein structure prediction, AF2 achieved remarkable performance when assessed on the individual domains of target protein sequences[1]. Because deep learning is a data-driven approach, two key factors contributing to the success of AF2 are the completeness of the structural space of single-domain proteins and an abundance of sequences in sequence databases[3]. Together, these factors have made it possible to train sophisticated neural network models for accurate protein structure prediction.

In addition to single-domain targets, AF2 also delivered excellent results on multidomain proteins[1] and has been applied to such proteins in several model organisms[2]. Considering that intra-protein domain-domain interactions are not physically different from interprotein interactions, these AF2 results are very intriguing. They hint that, in principle, AF2 could be repurposed to predict whether a pair of proteins interact and if so, to predict the quaternary structure of the resulting protein complex. After all, many proteins that form complexes in prokaryotes are fused into long, single-chain, multidomain proteins in eukaryotes[4]. The same physical forces that drive protein folding are also responsible for protein-protein associations[5]. Moreover, it is known that the structural space of protein-protein interfaces, the regions where direct physical contacts are found between protein partners forming a complex, is quite small[6]. Taken together, it is very likely that the neural network models trained for single-chain structure prediction have already learned the representations necessary to model protein complexes made of multiple single-chain proteins[3]. Indeed, some examples of using such a neural network model to predict complex structures were demonstrated with RoseTTAFold[7], whose design was inspired by AF2, even though its examples were likely present in its own model training set.

Until now, conventional approaches for predicting the structure of protein complexes include a docking component[8–11], which is limited by force-field accuracy and the completeness of conformational space sampling. A second class of approaches is template-based methods, which utilize experimentally resolved multimeric structures[12–16]. More recent methods incorporate inter-protein residue-residue contact predictions obtained via co-evolutionary analysis[17] and a deep learning algorithm[18].

Can AF2 be adapted to predict the structure of a protein complex? After the release of AF2, efforts immediately began to seek an answer. One early such study involved simply connecting two protein sequences with a poly-glycine linker, thus converting it into a single-chain structure prediction problem[19]. A much better solution is to modify the "residue_index" feature used by AF2, which eliminates the need for a covalent linker that likely creates artifacts[20,21]. Meanwhile, studies have been carried out whereby models of single proteins generated with AF2 are used with docking methods. They are based on the idea that AF2 generates high-quality monomeric models that could improve the chance of native-like poses in docking[22]. One issue with these studies, as some authors pointed out, is that the benchmark set tested includes protein structures used to train the AF2 deep learning models. Although the AF2 models were not trained on protein complex structures, the use of the holo monomers in training compromises rigor because AF2 likely provides an "observed" holo-structure for docking.

Going beyond the prediction of the structure of the protein complex given that the proteins interact, are more fundamental but more challenging questions: Can AF2 be adapted to predict protein-protein interactions given an arbitrary pair of protein sequences, and more generally, can it identify higher-order protein complexes? Several high-throughput experimental techniques

have been designed to identify interacting protein partners[23–26], but their results are far from complete and are often mutually inconsistent. Computationally, template-based approaches have been used[27], but they are limited to the detection of homologs. Combining standard protein-protein docking methods with co-evolutionary signals[28] or even with deep learning models[29] has also been carried out on full proteomes. These are powerful approaches, but they rely on paired multiple sequence alignments (MSAs) as inputs. Generating paired MSAs requires the identification of orthologous sequences across species, which is impractical in many cases because it is confounded by the presence of paralogs in eukaryotes, protein cross-talk in disease pathways, and pathogen-host interactions. After all, one main consequence of evolution is the diversification of protein functions by producing paralogs[30]. These paralogs may interact with different partners without using a conserved interaction mode. Therefore, it is highly desirable to develop an approach that is not dependent on paired multiple sequence alignments.

In this study, using multiple test sets and without using paired sequence alignments, we demonstrate that AF2 can be adapted to predict both the presence of protein-protein interactions and the corresponding quaternary structures. Although our tests are primarily conducted on dimers, the method, AF2Complex, can be applied to higher-order oligomers, and we show examples of such. Critically, it is necessary to devise proper metrics to estimate the confidence of a predicted protein complex model. By adapting metrics previously introduced for comparing the similarity of protein-protein interfaces[31], we introduce metrics for assessing the likelihood of protein-protein interactions. When AF2Complex was applied to a previously defined "gold standard" interaction set in *E. coli*[32], it found that many protein pairs are likely due to associations in large assemblies that are not necessarily in direct physical contact. Finally, we apply AF2Complex to make predictions on sought-after assemblies of a cytochrome *c* biogenesis system[33,34].

## Results

**AF2Complex overview**. The design of AF2Complex is illustrated in Fig. 1 with the details in Methods. Given query sequences of a target protein complex, the input features for each query are first collected by applying the original AF2 data pipeline. Then, AF2Complex assembles the individual monomer features for complex structure prediction. Among the input features, the most critical are the MSAs, which are obtained by extending each monomeric alignment sequence to the full complex length with gap paddings. Correspondingly, to mark separate peptide chains we sequentially increase the residue index feature of the second or later monomer(s) by a large number. The structural templates of monomer sequences are also re-indexed accordingly. If the input contains multiple copies of the same sequence, i.e., a homo-oligomer, it is treated as if they were heterogeneous sequences. In this way, one can readily reuse pre-computed features for individual sequences, e.g., from proteomes of species, for protein-protein interaction screening without any extra step such as MSA pairing. The input features for the putative complex are then separately supplied to AF2 deep learning (DL) models, and the resulting structure models retained for analysis. Finally, the likelihood of complex formation is assessed by two metrics: the interface-score and the predicted interface TM-score (piTM), both of which evaluate the confidence of the predicted protein-protein interface if found in an assessed model. Each of these two scores ranges from 0 to 1, where a higher score indicates higher confidence.

We note that AF2Complex was originally based on the five monomer DL models of AF2. After the release of AF-Multimer[35], additionally AF2Complex was adapted to take five multimer DL

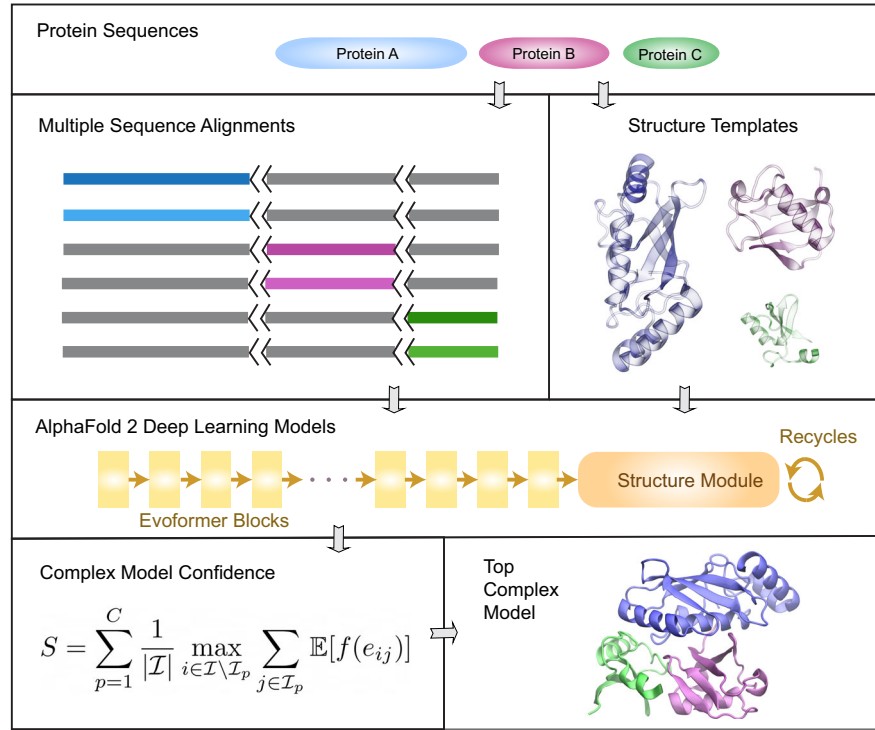

**Fig. 1 Overview of the AF2Complex workflow.** The multiple sequence alignments of query protein sequences A (blue), B (purple), and C (green) are joined together by padding gaps (grey) in the MSA regions belonging to other proteins, and the short black lines represent an increase in the residue index to distinguish separate protein chains. Structure templates for individual proteins are also retrieved from the Protein Data Bank. Using these sequence and template features, an AF2 DL model generates a complex model after multiple recycles. The interface residues between proteins in the final complex model are then identified and their interface-score $S$ is calculated to rank model confidence.

models for predictions using unpaired MSAs. Throughout this work, without note, we present structural models obtained with AF2's monomer DL models, rather than with AF-Multimer models.

**Accurate predictions on some CASP14 multimeric targets**. We first applied AF2Complex to the multimeric targets of CASP14[36,37]. To simulate a CASP14 prediction scenario, the input feature predictions described below were obtained by using databases available prior to May 15, 2020, CASP14's starting date. Because the structures of many assembly targets have not been released to the public, it is impossible to conduct a reliable statistical analysis of our predictions. However, they do showcase the potential of AF2Complex.

Figure 2 displays the results of AF2Complex on some challenging targets. The first example, H1072, is a heterotetramer (stoichiometry $A_2B_2$) consisting of two copies of two coiled-coil protein sequences[38]. Despite the simple topologies and the availability of an experimental structure for one monomer, H1072 is a difficult target. No group participating in the CASP14 competition provided a correct model of the complex[36]. In contrast, the top model by AF2Complex achieved a remarkable TM-score[39] of 0.90 when superimposed onto the experimental structure (Fig. 2a). The second example, H1065, is a challenging heterodimer as one component lacks a homolog in the PDB[36]. In this case, AF2Complex generated a highly accurate complex model with a TM-score of 0.94 (Fig. 2b). The interface similarity score (IS-score), which was designed to evaluate dimeric protein-protein interfaces[40], has a high value at 0.60[20]. The third example, T1070o, is a homo-trimer with intertwined β-sheets at the N-termini (Fig. 2c). Although we could not evaluate the overall complex structure because its experimental structure is unavailable, we were able to evaluate a monomeric structure that contains a free-modeling domain target (T1070-D1). If we extract

this monomeric domain from our top complex model and compare it to the native structure, our model for T1070-D1 yields a TM-score of 0.74, which is a significant improvement over 0.62 by AF2 in its official CASP14 assessment. This example indicates that, by modeling the entire homo-oligomeric target complex, one may obtain a structural model with higher quality, especially for an intertwined oligomer.

The last two targets are from large molecular machines whose full experimental coordinates are unavailable. One of them, H1060, is part of a T5 phage tail assembly[36,37]. The total size of this complex target is 6582 residues, which is too large to be modeled in full. However, it is possible to model its four ring-like substructures, which are formed by 3–12 copies of four distinct monomers. AF2Complex returns models of these four rings at highly confident interface scores ranging from 0.56 to 0.83 (Fig. 2d). The most challenging one is the 12-membered ring, for which our model forms an ellipse instead of the expected circular structure. Using AF-Multimer DL models, we obtained a single model with the expected C12 symmetry using unpaired MSAs but failed to build a physical model with paired MSAs (Supplementary Fig. 1). The last example, H1097, is a DNA-directed RNA polymerase from *Bacillus* phage AR9. It is composed of five hetero-monomers, totaling 2682 amino acids. AF2Complex generates a highly confident model with an interface score of 0.79 (Fig. 2e). Given that there are quite a few RNA polymerase structures in the current PDB, perhaps this result is not surprising. But the fact that AF2Complex can produce a model without paired MSAs in this case strongly indicates that paired MSAs might not be essential.

**Significantly higher accuracy over docking-centric approaches**. Next, we conducted a benchmark test using 17 heterodimers released after Apr 30, 2018, which was the cutoff date of PDB

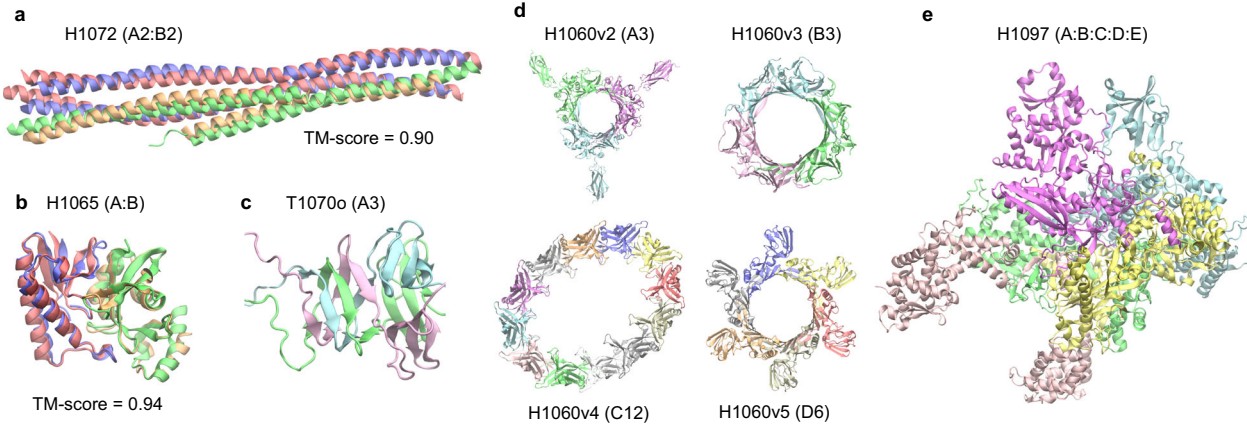

**Fig. 2 Top complex models generated by AF2Complex for selected CASP14 assembly targets.** Each target is labeled with its target name, e.g., H1072, followed by its stoichiometry in parentheses, e.g., $A_2:B_2$. For targets with available experimental structure coordinates, the similarity between the model and experimental structure is assessed by the TM-score[39]. For other structures only an image of the predicted model is given. Models are colored red and green, and experimental structures are in blue and gold. **a** SYCE2-TEX12 delta-Ctip complex. **b** N4-cytosine methyltransferase. **c** G3M192 from *Escherichia virus CBA120*. Only the N-terminal domains, which have an intertwined complex structure, are shown from a model of the full trimer. **d** Four rings from the T5 phage tail subcomplex. **e** DNA-directed RNA polymerase from *Bacillus phage AR9*. All images were generated with the program VMD[66].

structures collected for training the AF2 models. This set, named CP17, was curated for assessing various docking-centric strategies in a recent study[22]. One such strategy is to build a complex model using the ColabFold version of AF2[20], then split the monomers from the predicted AF complex and use the ClusPro[41] docking method to generate complex models. This strategy yields an acceptable or better top-ranked model for fewer than half of the targets. By comparison, the overall top models from AF2Complex are acceptable or better in 15 of 17 (88%) cases, and 13 (76%) models are of medium or high quality according to the DockQ score[42] (Fig. 3a, Supplementary Data 1), with a mean of 0.62, a dramatic improvement from 0.25 of the docking-centric approach. Another complex, docking-centric strategy increases the mean of DockQ score from 0.25 to 0.47[22]. Nevertheless, the improvement still falls behind AF2Complex on the same set (Fig. 3b). However, in one of the two cases when AF2Complex failed, the combined strategy resulted in a high-quality model. In these two failed cases, each has one monomer with only single-digit depth in its monomeric MSAs, which may explain these failures.

**Improvement on AF-Multimer.** Moreover, using AF2 monomer DL models, AF2Complex performs quite close to AF-Multimer[35] on the CP17 set (Fig. 3c), with a mean DockQ score of 0.62 versus 0.65. The mean score of AF2Complex increases to 0.74 using AF-Multimer DL models (Fig. 3d), thanks to unpaired MSAs, increased recycles and the interface-score metric (Supplementary Data 2). By combining the overall best model from AF2Complex runs using either monomer and multimer DL models of AF, we obtain acceptable or better models for all 17 targets.

To address the small target size of CP17, a large-scale benchmark study was further carried out on 1,192 dimers and 593 oligomers. On 440 heterodimers obtained using the same DL models, AF2Complex performs significantly better than AF-Multimer, albeit at smaller advantage (median/mean 0.69/0.56 versus 0.65/0.55, $p$ value $= 5 \times 10^{-3}$, Wilcoxon signed-rank test, paired, one-tailed, $n = 440$, see Supplementary Text and Supplementary Fig. 2), whereas their performance on the homodimer set is comparable. AF2Complex further improves (median/mean 0.70/0.57 on the heterodimers, $p$-value $= 6 \times 10^{-4}$, same Wilcoxon test as above) if the overall top-ranked model of both monomer and multimer DL runs are assessed for the

heterodimers. It must be pointed that AF-Multimer (version 2.1.1) may yield unphysical models with severe clashes for a complex, especially for large oligomeric targets, including homo-oligomers. In contrast, AF2Complex mitigates this issue with unpaired MSAs using the same multimer DL models and deals much more effectively with unpaired MSAs and the original monomer DL models of AF2 (Supplementary Fig. 3).

**Predicting interactions among arbitrary protein pairs.** Having been rather successful, an obviously tantalizing question is: Can this approach be applied to predict direct protein-protein interactions for an arbitrary pair of proteins? To answer this question, we devised a test using the 34 unique protein sequences from the CP17 set. The goal was to find the 17 true interacting pairs given in CP17 from the 561 all-against-all pairwise combinations. Here, we naively assumed that all protein pairs other than the CP17 pairs are non-interacting, and any hit above a cutoff value of a metric adopted for evaluation is a false positive. Figure 4 shows the results by using four different metrics to predict protein-protein interactions. Note that the model predictions for all pairs were carried out under exactly the same configuration in AF2Complex runs. Overall, both the interface-score and the piTM-score demonstrate a clear advantage over the other two metrics, the pTM-score and pLDDT-score of AF2[1]. Because we expect that most pairs of proteins are non-interacting, we focus on the regime of low false-positive rate (i.e., FP < 0.1) in the receiver operating characteristic (ROC) curve. The normalized area under the curve (AUC) of this plot, $AUC_{0.1}$, is 0.72 and 0.69 for interface-score and piTM, versus 0.49 and 0.10 for pTM and pLDDT, respectively. For reference, random guessing yields an $AUC_{0.1}$ of 0.05. As expected, pLDDT is not ideal for evaluating protein complex models because it was designed for single domain evaluation. Although the pTM metric is much more discriminating than pLDDT, it is still much worse than interface-score or piTM in this regard. The same trend is also displayed in the precision-recall plot, whereby we achieved ~45% recall (equivalent to the true positive rate) at ~90% precision, and the recall increases to ~70% at ~45% precision. Correspondingly, the interface-score and piTM values are 0.55/0.59 and 0.45/0.50, respectively. Overall, the results encouragingly conclude that AF2Complex can be used to predict protein-protein interactions.

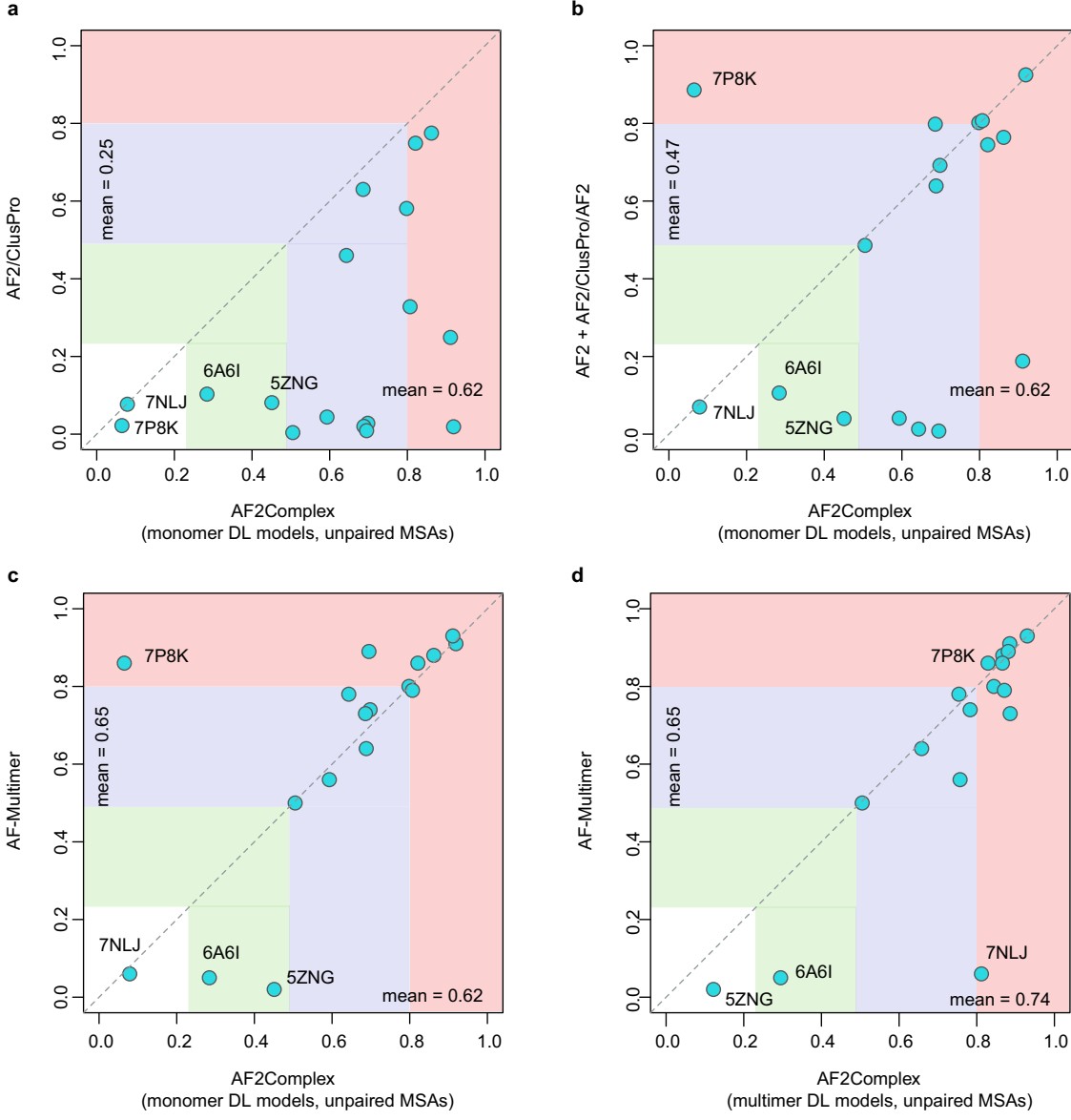

**Fig. 3 Comparison of AF2Complex and three alternative approaches on the CP17 set.** The coordinates of the circles correspond to the DockQ scores[42] of the top overall models from each approach versus AF2Complex. **a** AF2 models docked by ClusPro[22]. **b** Docking models refined by AF2, plus additional complex models obtained by running AF2 on paired MSAs according to Ref. [22]. (**c**, **d**) AlphaFold-Multimer[35]. The AF2 deep learning models trained for the prediction of monomeric protein structures, denoted as "monomer DL models", were employed by AF2Complex in (A–C), and the AF-Multimer deep learning models, denoted as "multimer DL models", were applied with AF2Complex in (**d**). All MSA inputs to AF2Complex are unpaired as described in Methods. Vertical and horizontal blocks represent the regions of incorrect (white), acceptable (green), medium (blue), and high-quality (red) complex models according to the DockQ score. The four most challenging targets are marked by their four-letter PDB accession codes.

**Application to the *E. coli* proteome**. The *E. coli* proteome consists of about 4,400 protein sequences. An all-against-all run with AF2Complex would require about 0.5 to 2.4 million computing node hours on the Summit supercomputer, which is beyond our allocation. Instead, we focused on a "gold standard" set of 701 PPIs previously curated largely from high-throughput experiments, and a set of 6,849 randomly selected, putatively non-interacting pairs. Here, we have two goals: one is to test AF2Complex on a large scale; the other is to build complex models for some known interacting protein pairs whose structures are difficult to determine experimentally. We speculated that the "gold standard" set contains pairs from a large assembly that are not necessarily in direct contact, e.g., as in a ribosome. To test this hypothesis, we divided the positive set into subsets, whereby each monomer in the subset does not appear more than

$C$ times in the putative interacting set (the full set is covered when $C < \infty$). Figure 5 shows the corresponding ROC and PR curves for these sets. The ROC curve displays a clear trend in which higher $C$ values correspond to lower true positive rate or recall. This analysis suggests that some of the pairs in the positive set do not interact directly, yielding low or even zero scores. When we considered proteins that appear only once (i.e., $C = 1$), we obtained a result that largely recapitulates the benchmark performed above, with a slightly lower $AUC_{0.1}$ of 0.60. The $AUC_{0.1}$ drops to 0.50 at $C = 3$, 0.40 at $C = 5$, and 0.22 for the full set, likely due to the inclusion of more non-direct interacting pairs as $C$ increases. Further analysis also found a barrier to accurate modeling is the lack of the exact context of the protein-protein interaction. For example, the chaperonin protein GroL is the most frequent monomer in this set, appears in 79 pairs. It is part

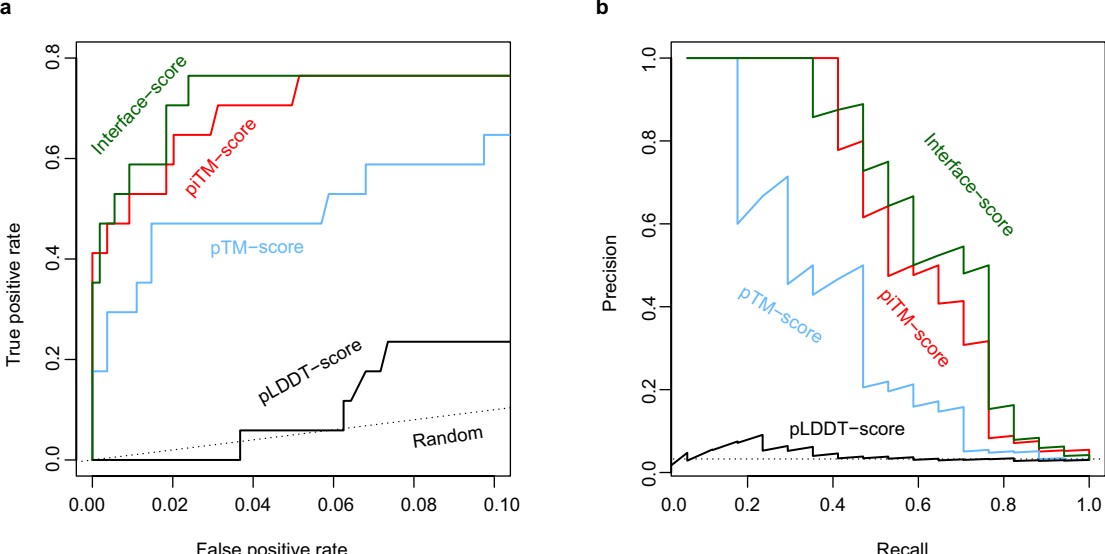

**Fig. 4 Identification of true interacting protein pairs in the all-against-all pool for the CP17 set by various confidence metrics. a** Receiver operating characteristic curve and (**b**) the precision-recall curve. The random curve is the expected result by randomly guessing interacting protein pairs. piTM-, pTM- and pLDDT-score denote predicted interface TM-score, predicted TM-score, and predicted local distance difference test score, respectively.

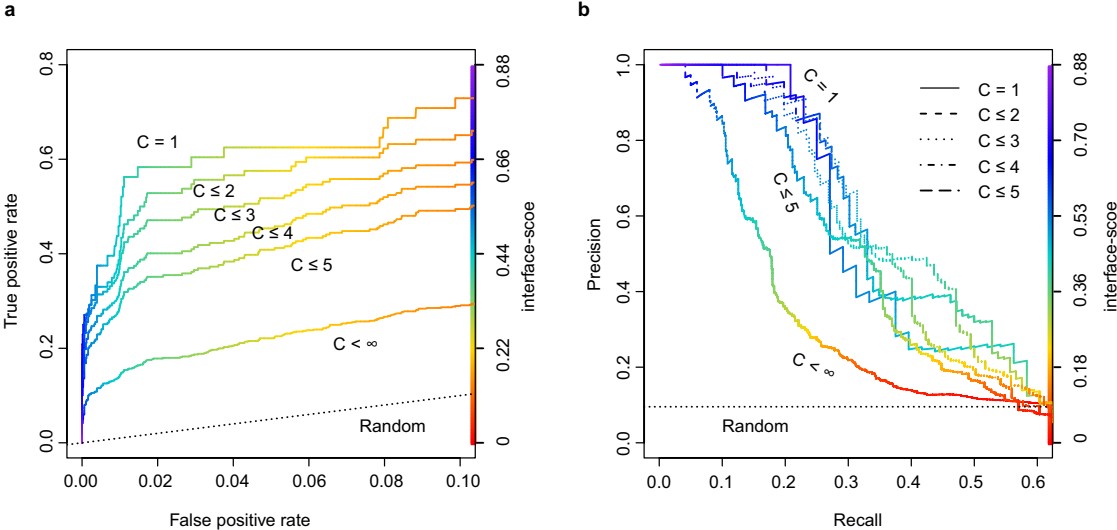

**Fig. 5 A large-scale test on the *E. coli* proteome suggests that many pairs previously thought to interact directly are likely in assemblies of components that are not necessarily in direct contact.** The interface-score was used as the varying metric to derive the (**a**) ROC curve and (**b**) the precision-recall curve. For a dimer target, $C$ is defined by the maximum of the appearances of its two monomers in this data set.

of a large assembly that requires seven GroL copies forming a ring stacked with another heptameric GroE ring. However, when GroL was modeled alone with another putative partner, we only found models with low confidence scores. Despite this difficulty, the result suggests that AF2Complex performs as expected in a large-scale test. Of the positive set, among the predicted models with confident scores (interface-score > 0.45), about 40% of these predictions have not been experimentally characterized (defined if both monomers share >70% sequence identity with sequences found in the same PDB entry). Therefore, interesting discoveries are expected from these models.

**Cytochrome *c* biogenesis system I.** Two *E. coli* targets with high-confidence models, CcmE/CcmF and CcmF/CcmH, caught our attention. These proteins belong to the cytochrome *c* maturation

(Ccm) system I, which is composed of eight constituents (CcmABCDEFGH)[33,34]. We note that *E. coli* CcmH has a fused C-terminal domain that appears as the standalone protein CcmI in other species with a similar Ccm system. As illustrated in Fig. 6a, it is thought that the Ccm system I consists of two modules: module 1 includes CcmABCD and is responsible for acquiring and loading a heme molecule onto the heme chaperone CcmE[43]. CcmE then shuttles the heme from module 1 to module 2, composed of CcmFGH, where the heme is delivered to CcmF[44]. Subsequently, CcmFGH covalently attaches the heme to nascent cytochrome *c*-type proteins[45,46].

Many mechanistic details of the Ccm system are still unclear, in part because there are no structures of the assembled modules. To date, the best effort is a partial model of the CcmCDE complex generated using co-evolutionary analysis[47]. The main reason for this knowledge gap is that the assemblies involve transient but

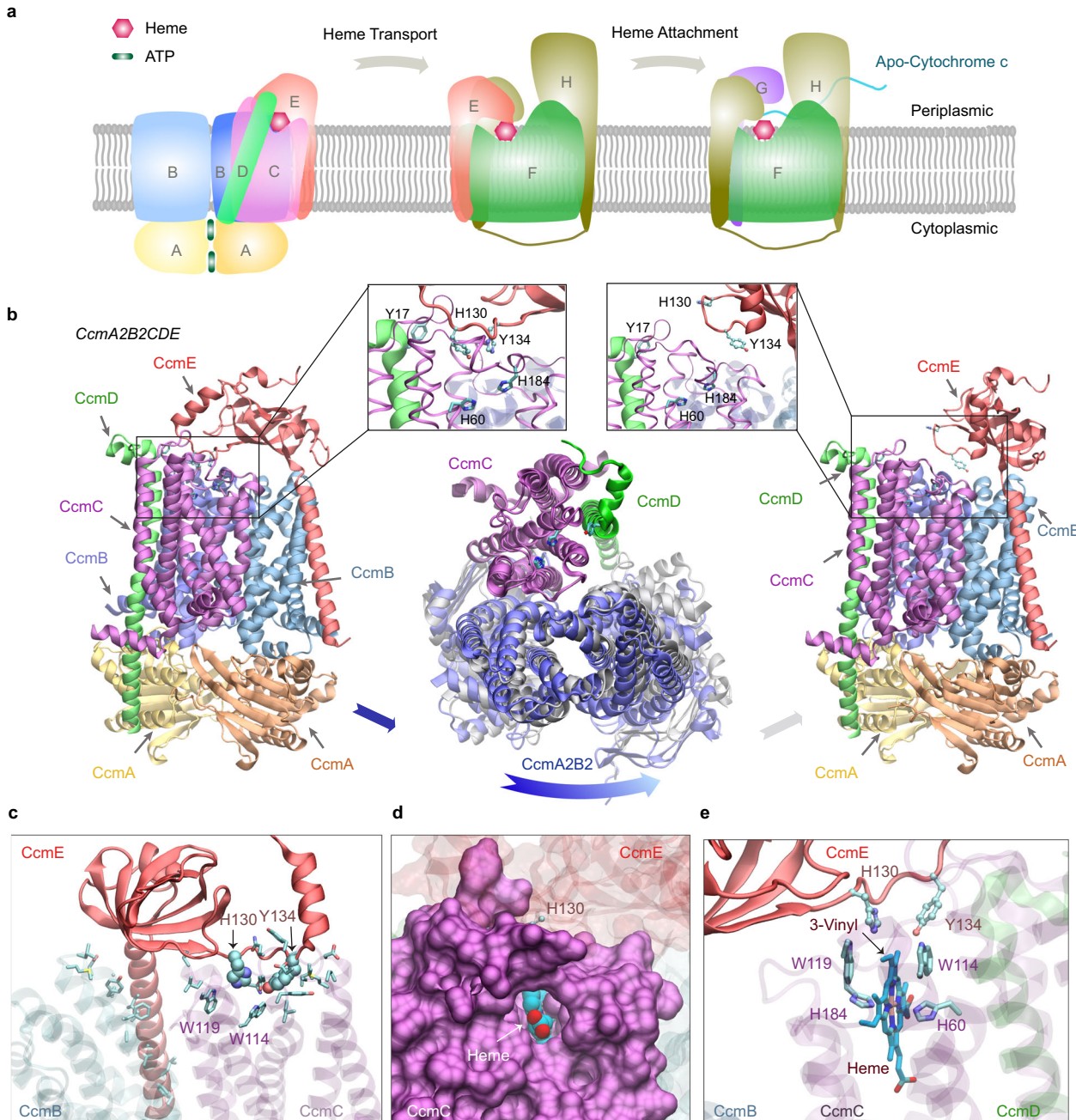

**Fig. 6 *E. coli* cytochrome *c* maturation system I. a** An illustration of the Ccm I system, composed of eight proteins named CcmABCDEFGH. The system covalently attaches heme molecules to cytochrome *c* proteins via three functional complexes. **b** Two models (left and right panels) of one complex: CcmA$_2$B$_2$CD engage CcmE (left panel) and disengage CcmE (right panel). which loads a heme from CcmA$_2$B$_2$CD and chaperones it to CcmF. Insets show conserved residues implicated for heme binding in CcmC, CcmD, and CcmE, respectively. Conformational differences between these two models are shown in the middle panel, where the backbone of CcmC was used to superimpose the two models. Viewed from the periplasmic side, the two conformations of CcmA$_2$B$_2$ are displayed in blue and grey. Movement relative to CcmC is evident in CcmA$_2$B$_2$ but not in CcmD. For clarity, CcmE is omitted in this superposition plot. **c** A view of interactions between CcmCD and CcmE in their engaged structural model shown in the left panel of (**a**). CcmCD representations are transparent for clarity. The side chains of interacting residues are shown. His130 and Tyr134 of CcmE are shown in the van der Waals representation, and other interacting residues, including Trp114 and Trp119 from the heme-binding WWD domain of CcmC, are shown in the licorice representation. (**d, e**) Views of a heme molecule bound in the putative binding-pocket in CcmC, using the structural model in which CcmE is bound to CcmA$_2$B$_2$CD as the initial apo structure. A pore for heme access in CcmC manifests, where CcmC is rendered in a surface representation (**d**). The heme is displayed in van der Waals (**d**) and licorice (**e**) representations. The vinyl group expected for His130$^{CcmE}$ attachment is marked in (**e**).

essential interactions among membrane proteins (except for CcmA) that are difficult to capture experimentally. We sought to address this knowledge gap with AF2Complex by modeling about two dozen combinations of Ccm components. From these computational experiments, we present the confident models of

three assemblies, A$_2$B$_2$CDE, EFH, and FGH, the last one with and without apocyt peptides, all with high interface-score (0.82, 0.56, and ~0.72, respectively). These models are likely biologically relevant, and we were able to model the heme in the expected sites in all these models.

**CcmA₂B₂CDE**. First, we identified two different conformational states in the top two highest-ranking models of A₂B₂CDE (Fig. 6b). Extensive interactions are present between CcmCD and CcmE in one model (Fig. 6b left panel), where the heme chaperone CcmE extends its heme-binding loop, characterized by two conserved residues His130$^{CcmE}$ and Tyr134$^{CcmE}$, to reach the heme-binding site (HBS) in CcmC that includes His60$^{CcmC}$ and His184$^{CcmC}$, and another conserved residue, Tyr17 of CcmD. The heme-binding loop of CcmE is oriented away from the HBS of CcmCD in the other model (Fig. 6b right panel), in which the interactions between CcmC and E disappear, and the only remaining inter-protein residue-residue contacts are between a pair of transmembrane helices of CcmC and CcmE. This conformation presumably corresponds to a resting state before heme loading, after heme unloading, or both. The disengagement of CcmE leads to conformational changes within CcmA₂B₂CD (Fig. 6b center panel). Using CcmC as the reference to superimpose the two complexes, movement is evident in CcmA₂B₂ but not in CcmD, which appears tightly coupled with CcmC. The root mean square deviation (RMSD) of the CcmC backbone is ~1 Å. Corresponding to this movement, the number of residue-residue contacts between CcmB and CcmC drops by 45%, and the CcmBC protein-protein interfaces display larger changes between the two models than the other interfaces (similarity score, IS-score, of 0.65 versus scores >0.84). These large conformational changes could be the result of ATP hydrolysis within CcmA. As previously proposed[43], the energy of the hydrolysis could be harnessed to release the cargo-loaded CcmE.

Extensive contacts are present in the model with CcmC tightly bound with CcmE. The interactions involve Trp114 and Trp119 of a tryptophan-rich loop (WWD domain[48]), sitting at one edge of the binding pocket of the heme (Fig. 6c). A pore is visible between transmembrane helices 2 and 5 of CcmC and exposes a heme molecule bound to CcmC (Fig. 6d). The pore may allow the access of heme from the outer leaflet, but no channel for potential heme trafficking is present within CcmC, as previously speculated[48]. The role of His60$^{CcmC}$ and His184$^{CcmC}$, predicted heme iron-coordinating residues, is confirmed as well (Fig. 6e). His130$^{CcmE}$ is only ~4 Å away from the 3-vinyl of heme (IUPAC numbering), which is the proposed site of covalent attachment to His130$^{CcmE}$ to complete the heme delivery to CcmF.

**CcmEFH**. Next, we addressed the question of how a heme-carrying CcmE could deliver heme to the CcmFGH complex. We obtained a confident model in which CcmE is in complex with CcmF and CcmH (Fig. 7a). CcmE interacts with CcmF such that the HBS of CcmF faces the HBS of CcmE, which has a heme-handling motif like that in CcmC. The distance between His130$^{CcmE}$ and His303$^{CcmF}$ is ~6 Å (Fig. 7a inset). CcmF has two heme-binding sites: one (P-heme) for the cytochrome attachment and the other accessory heme (TM-heme) that assists in the ejection of the P-heme[34]. His303 is highly conserved and is known to coordinate the Fe center of P-heme. Similar interacting poses between CcmE and CcmF were obtained in models generated in the absence of CcmH.

Consistently, the heme-bound model displays the P-heme coordination role of His303$^{CcmF}$ and His173$^{CcmF}$, and the TM-heme is coordinated by His261$^{CcmF}$ and His461$^{CcmF}$ (Fig. 7b). TM-heme was co-crystallized with CcmF from *Thermus thermophilus* (*Tt*CcmF) in a recent published X-ray structure[49]. Without using this structure as a template, our predicted CcmF model has a TM-score of 0.92 compared to the experimental structure. However, a critical loop containing the equivalent histidine of *E. coli* His303$^{CcmF}$ is missing in the *Tt*CcmF structure, suggesting that the P-heme binding pocket might be flexible. Indeed, only

one of the two expected tryptophans (W229 and W236) from the WWD domain is in contact with the P-heme in the heme-bound model. Moreover, CcmF in our model does not expose a pore as observed in the *Tt*CcmF structure, in which the pore was speculated to enable P-heme access[49]. The structural difference may be attributed to a lipid molecule occupying the pore in the crystal structure but absent in our computational model.

**CcmFGH**. After the heme is delivered to CcmF, the final step performed by this system is the attachment of the heme to apocytochrome *c* (apocyt *c*). This step involves a complicated mechanism that is not fully understood[46]. However, our model of the CcmFGH complex provides structural insights into the mechanism (Fig. 8a). First, we note the mobility of the N-terminus of CcmH (which would be the full CcmH in many other systems that also have CcmI but is fused to CcmH in *E. coli*). In the absence of CcmE, the N-terminus of CcmH occupies the site otherwise occupied by CcmE, essentially moving closer to the HBS of CcmF. This configuration leaves an opening for CcmG, another thiol-disulfide oxidoreductase (like CcmH), now sitting at the site previously occupied by the CcmH N-terminal domain. Remarkably, the CcmFGH complex is arranged such that a reaction groove is formed in which an apocyt *c* can be sequentially passed among the CXXC motifs of CcmG (Cys80 and Cys83) and CcmH (Cys43 and Cys46) to reach the HBS of CcmE (Fig. 8a).

To investigate further, we modeled CcmFGH together with 11 different apocyt peptides, each containing one or two CXXCH motifs. Remarkably, in all top models, the apocyts are invariably located in the predicted reaction groove (Fig. 8b). Physical contact between Cys46$^{CcmH}$ and an apocyt cysteine is present in some models (Fig. 8c). We further modeled a heme in the expected P-heme site of CcmF. The heme-bound model is largely as expected, but more conformational changes are necessary for apocyt-heme attachment, as the cysteines of the apocyt are still ~16 Å away from the 8-vinyl group of the P-heme. The speculation echoes large conformational changes upon heme-binding revealed in recently determined structures of CcsBA, a cytochrome *c* synthetase from a different Ccm system[50].

Of the previously proposed mechanisms[46], our model supports the following: after entering the reaction groove, apocyt *c* is first reduced by Cys80 and Cys83 of CcmG. Next, the reduced apocyt attacks one of Cys43 and Cys46 of CcmH to form a mixed disulfide. This intermediate complex then retrieves the heme acquired by CcmF, and subsequently the mixed disulfide is resolved by the second cysteine of CcmH. Finally, the holo-cytochrome *c* is released. The CXXC motif of CcmH then reverts to the oxidized state, and CcmG dissociates to be reduced by the thiol-disulfide interchange protein DsbD. In all models of CcmFGH and also models from separate modeling of a CcmGH complex, the CXXC motifs of CcmG and H are separated by ~15 Å, and hence the reduction of a mixed disulfide between these two motifs as proposed in an alternate mechanism[46] is unlikely according to these models. Interestingly, the two CcmH domains, encoded in two ORFs in some organisms but fused together in *E. coli*, are linked by a long loop without direct interactions. This is an exception to the notion that fused proteins directly interact[4]. In this case, both CcmH domains instead interact with a third protein, CcmE. The function of the CcmH C-terminal domain remains unclear, but likely involves interactions with an apocyt *c*.

## Discussion

Our findings clearly demonstrate that AF2 can be adapted to predict the structure of protein complexes at much higher accuracy

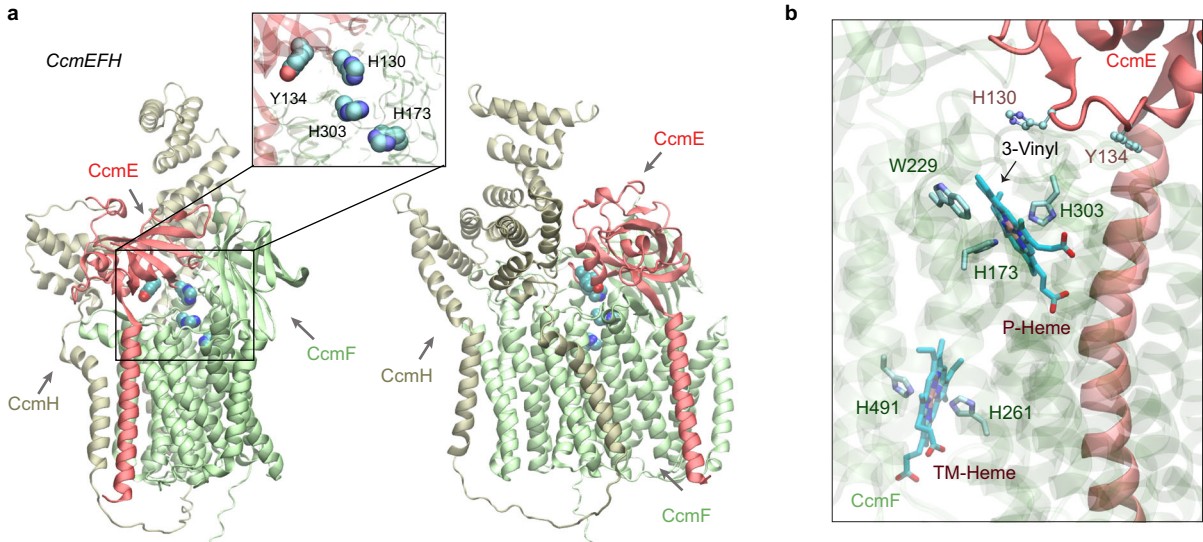

**Fig. 7 Structural models of the CcmEFH complex from *E. coli*. a** CcmE is believed to deliver heme to CcmF. Two views of a top model generated by AF2Complex are shown in the cartoon representation. The inset shows the key heme handling residues, His130 and Tyr134 of CcmE and the two histidines of CcmF. **b** Two heme molecules computationally placed in the expected heme-binding sites of CcmF using the model shown in (**a**). The critical heme Fe-coordinating residues, His173 (P-His1) and His303 (P-His2) for the P-heme delivered by CcmE and eventually attached to an apo-cytochrome *c* protein and His261 (TM-His1) and His461 (TM-His2) for the cofactor TM-heme are also shown.

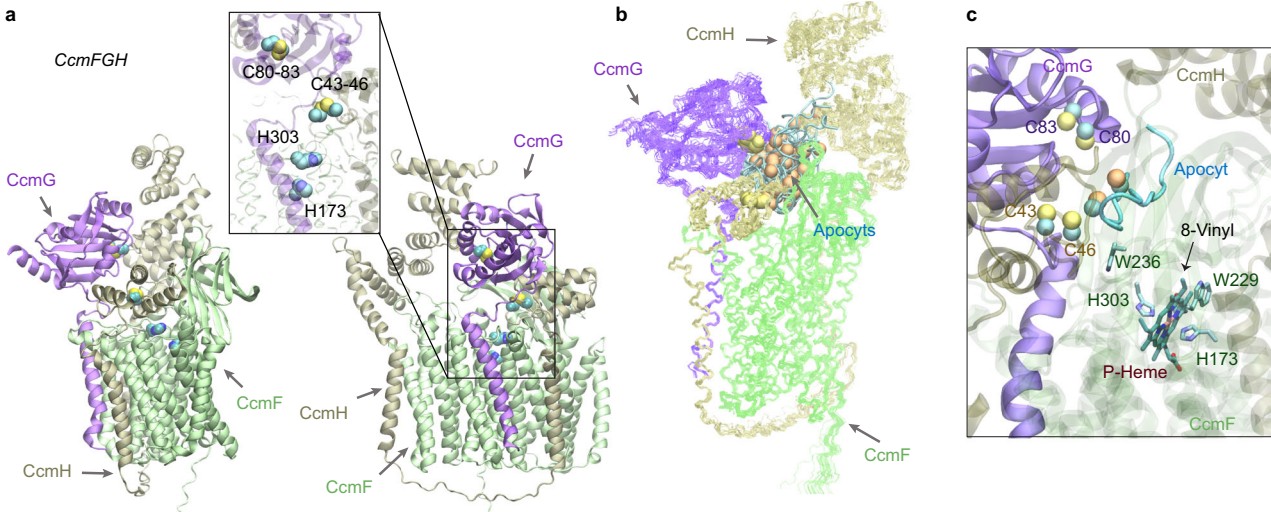

**Fig. 8 Structural models of the CcmFGH complex with and without apo-cytochrome c substrates (apocyts) from *E. coli*. a** Two views of a top model of CcmFGH are shown in the same orientations as the two views in Fig. 7a, respectively. The CcmH N-terminal domain moves closer to the heme-binding sites of CcmF, leaving space to accommodate CcmG that now binds CcmF with CcmH. Critical cysteines of the CXXC motifs of CcmG (Cys80 and Cys83) and H (Cys43 and Cys46), and the P-heme-binding histidines (His173 and His303) of CcmE are shown in the vdW representations in the inset. **b** Superposition of 22 AF2Complex models of the CcmEFH and apocyt acceptors. CcmFGH complexes are shown in lines, and apocyts are shown as cyan tubes. All apocyts are found within the same groove formed by the three Ccm proteins. The superposition used the backbone atoms of CcmFGH as the reference. The sulfur atoms from the CXXC motifs of apocyts are shown in orange spheres to differentiate from those of CcmGH. **c** A heme molecule computationally docked to the P-heme binding site in CcmF using one of the models shown in (**b**). One of the Cys residues of apocyt is found within 4 Å from Cys46 of CcmH. The distance between the other Cys residue of apocyt and the 8-vinyl group of the heme is about 16 Å.

than classical docking approaches, even if the docking approaches use monomeric structures predicted by AF2. One reason is that predicting all the protein structures involved in a complex simultaneously by AF2Complex may overcome issues associated with rigid-body docking. Importantly, we have shown in multiple benchmark tests that high-quality complex prediction can be achieved without using paired MSAs as input, which also significantly enhances AF-Multimer and returns more physical complex models. This feature may lower the barrier for applications

including some challenging cases in which pairing MSAs is impractical.

But why is it possible to achieve successful complex modeling without using paired MSAs? After all, for predicting single-chain structures and also for predicting complex structures, MSAs for each individual protein are still necessary and important. One reason is that sophisticated deep learning models reduces their reliance on large MSAs[1,51]. Moreover, we speculate that the accurate amino acids packing capabilities offered by AF2 deep

learning models may be another key reason[1,3]. In particular, the structural module of AF2 ignores the sequential order of amino acids and has likely learned energetically favorable patterns among packed amino acids. If these patterns are applicable universally to amino acids of either intra- or inter-proteins, success is then expected. Empirically, the results above indicate this is likely the case. After all, protein-protein interactions are not physicochemically different from what drives protein folding in the first place. Their interface structures likely have been learned during the training of AF2 deep learning models for monomeric protein model prediction.

Furthermore, by assessing the confidence of a predicted complex model with carefully designed metrics, one may generalize this deep learning approach to predict direct protein-protein interactions. We demonstrate that the interface-score or piTM metric can be applied to effectively predict interacting pairs. When applied to an *E. coli* proteome, we were able to infer that some of the previously selected interacting pairs in the "gold standard" set are likely from large complex assemblies without direct interactions.

Using the *E. coli* cytochrome maturation system I as an example, we demonstrated that this powerful computation tool can be applied to interrogate a molecular system comprised of many proteins arranged in different packing orientations. By way of example, AF2Complex can generate highly confident models that depict the complexes involved in the loading, release, and delivery of a heme-chaperone, including a reaction groove responsible for the final attachment of a heme to an apo-cytochrome protein. Remarkably, high-confidence models were obtained for these assemblies that include multiple conformational states involving transient interactions with the heme chaperone and apocytochrome *c* peptides. Although the approach is currently limited to structural models without directly incorporating a heme, docking studies suggest that our models are consistent with known biochemical evidence, though other conformations are also expected.

One major hurdle to this bottom-up approach for predicting protein interactions is that the context of such a hypothetical complex is often unavailable a priori. For example, if a complex involves one homodimer and another monomer, it would be difficult to model if we only consider a single heterodimer. Another challenge is post-translational modifications. For instance, proper modeling of CcmEFH and CcmFHI requires the cleavage of the N-terminal signal peptide of CcmH to obtain biologically accurate models. Nevertheless, the power of a deep learning-based approach for predicting direct protein-protein interactions has been demonstrated. It is expected to contribute profound structural insights into the understanding of many biological molecular systems.

## Methods

**AF2Complex workflow.** AF2Complex was initially built upon the official release of the source code and monomer neural network models of AlphaFold2 (version 2.0.1)[1], and subsequently upgraded to support the multimer neural network models of AlphaFold-Multimer (AF version 2.1.1). For the purpose of large-scale applications, the original data pipeline was separated from the neural network inference. The split allows us to derive input features for individual protein sequences and then reuse them to assemble input features for subsequent complex predictions. We used different sets of sequence libraries[52–55] and Protein Data Bank (PDB)[56] releases to generate appropriate input features for different test sets, as described below.

To generate the MSAs for predicting complex structures made of $N$ distinct protein sequences, each with a length $L_i$ and a stoichiometry number $S_i$ ($i = 1\ldots N$), we apply Algorithm 1 to the MSAs of individual proteins. The application creates a new set of complex MSAs, whose length is the sum of all individual sequences including multiple copies in the case of homo-oligomers, and whose depth is the sum of the depths of all individual MSAs. The complex MSAs are primarily composed of gaps, except for the regions in which each individual target sequence has its own window of MSAs (see Fig. 1 for a schematic example).

**Algorithm 1.** Complex MSA Creation
```
def make_complex_msa (protein_msa_list):
```
1: $msa\_length \leftarrow \sum_{i=1}^{N} L_i \times S_i$ ▷ $N$ is the number of proteins, $L_i$ and $S_i$ are the length and stoichiometry of protein $i$
2: $msa\_depth \leftarrow \sum_{i=1}^{N} D_i \times S_i$ ▷ $D_i$ is the depth of the MSAs of protein $i$
3: $msa \leftarrow$ initialize_msa_with_gaps ($msa\_length$, $msa\_depth$)
4: $col\_start \leftarrow 0$
5: $row\_start \leftarrow 0$
6: **for** $i \leftarrow 1$ to $N$ **do**
7: **for** $j \leftarrow 1$ to $S_i$ **do**
8: $col\_end \leftarrow col\_start + L_i$
9: $row\_end \leftarrow row\_start + D_i$
10: $msa[row\_start: row\_end, col\_start: col\_end] \leftarrow msa_i$ ▷ $msa_i$ is the set of MSAs of protein $i$
11: $col\_start \leftarrow col\_end$
12: $row\_start \leftarrow row\_end$
13: **end for**
14: **end for**
15: **return** $msa$

When supplied to a monomer AF DL model, the "residue_index" input feature for the target complex was modified by increasing the residue indices of individual protein sequences by $b(p-1)$, where $b$ is an arbitrarily chosen number of 200, which satisfies the condition larger than the coverage of the relative positional encoding at 32 (that is, the sequential distance between two residue indices $|i-j| \leq 32$, see Algorithm 4 of the Supplementary Method of reference[1]), and $p$ denotes the index of each monomer starting from 1. Likewise, the template structures for individual proteins were also collected for the complex prediction. However, we did not specifically search for and supply a complex template for the target. Without a note, the neural network models used for subsequent structure prediction were the five monomer AF2 models, each with a fine-tuned head for predicting paired alignment errors, which allows the prediction of the TM-score (pTM)[1]. We took advantage of this head for deriving metrics for evaluating complex predictions (see below). In the tests described in this study, we increased the number of recycles up to 20, depending on sequence lengths; the maximum number of recycles for target sequences longer than 500 residues was progressively decreased to reduce computational costs. The recycle steps were stopped early if the backbone $C_\alpha$ distogram converged[20]. In practice, these options may be adjusted by end users.

**Metrics for complexation evaluation.** Previously, we introduced the interface TM-score (iTM-score) and interface-similarity score (IS-score) for measuring the structural similarity between protein-protein interfaces[31]. Both scores were introduced to deal with issues associated with the TM-score, which is not ideal for comparing structure similarity of protein complexes[31,57]. In this study, we used a similar concept but modified it accordingly for estimating the confidence of a predicted complex model. We first introduce the predicted interface TM-score, piTM,

$$piTM = \max_{i \in \mathscr{I}} \frac{1}{I} \sum_{j \in \mathscr{I}} \frac{1}{1 + \left[\langle e_{ij} \rangle / d_0(I)\right]^2} \quad (1)$$

where $\mathscr{I}$ is the set of interface residues observed in the predicted model structure, and the cardinality of $\mathscr{I}$ is the total number of interface residues $I \equiv |\mathscr{I}|$. Using the local reference frames of interface residue $i$, the predicted alignment error head of AF2 gives an estimated distance $\langle e_{ij} \rangle$ for interface residue $j$ from its position in the experimental structure[1]. The piTM score is the optimal rotation/translation that gives the best-estimated score, and $d_0(I)$ is a normalization factor given by,

$$d_0(I) = \begin{cases} 1.24\sqrt[3]{I - 15} - 1.8 & \text{if } I \geq 22 \\ 0.02I & \text{if } I < 22 \end{cases} \quad (2)$$

Note that we adjust the original formula of $d_0$ to better deal with the cases where a low number of contacts are observed. Furthermore, we define the interface-score $S$ as the follows,

$$S = \sum_{p=1}^{C} \frac{1}{I} \max_{i \in \mathscr{I} \setminus \mathscr{I}_p} \sum_{j \in \mathscr{I}_p} \frac{1}{1 + [\langle e_{ij} \rangle / d_0(I)]^2} \quad (3)$$

which is similar to piTM, but we now calculate a piTM score for each protein chain $p$ of the complex separately and then sum the scores. Each chain $p$ has an observed number of interface residues $\mathscr{I}_p$, and $\mathscr{I}$ is the union of $\mathscr{I}_p$. The optimal local reference for calculating the score for chain $p$ can only be selected by interface residues not belonging to chain $p$. An important difference between our metrics and the ipTM score introduced in ref. [35] is that we focus on interface residues (versus full chains[35]), which is the most relevant for our interaction predictions.

**CASP14 multimeric targets.** We modeled all assembly targets if the total size of an assembly is less than 3000 residues, which is a limit imposed by our available computing resources. The sequence libraries employed for our predictions are UniRef90 created in 2020-01[58], and the reduced BFD, MGnify, and Uniclust30 libraries[52–55] provided with the AF2 release. All these libraries are composed of

sequences available prior to CASP14. For template retrieval from the PDB, we restricted ourselves to structures released before the starting date of CASP14 as well. From the final predictions, we selected some difficult targets with promising results to present.

**CP17 set**. This set was taken from ref. [22] and consists of 17 heterodimer targets released after 2018-04-30, on which the experimental structures were collected from the PDB for training the AF2 neural network models. We used the same sequence library as above but restricted the structural templates from the PDB to those dated before the same cutoff date for the AF2 training structures. In the benchmark test of the 17 targets, three independent runs were carried out, and each generated five top models. We evaluate the overall top-ranked model of the 15 models according to the interface score. This protocol was used to evaluate both monomer and multimer AF neural network models. In the second benchmark test of identifying the 17 heterodimers out of all 561 pairs of these 34 individual protein sequences of the dimers, one single run was conducted for all pairs. Up to 20 recycles were allowed in each of these runs.

The top-ranked (by the interface-score) models were compared with their corresponding experimental structures with the programs IS-score[40] and DockQ[42]. Inaccurate, acceptable, medium, and high-quality models are defined by DockQ score regimes [0.0, 0.23], [0.23, 0.48], [0.48, 0.80], and [0.80, 1.0], respectively.

**Dimer1193 and Oligomer562 sets**. To create large benchmark sets for objective evaluations of the AF2 models, we curated two sets from the experimental structures released recently in the PDB. First, we retrieved the information of all 120,703 protein assemblies on Jan 7, 2022, from the PDB. They were split into two subsets, 92,047 complex structures (Earlier Set) released prior to Apr 30, 2018, and 24,853 assemblies (Later Set) released after that date, which is the cutoff date for the structure collection used to train the AlphaFold DL models including both monomer and multimer DL models[35]. We assume all assemblies in the Earlier Set were used for DL training and removed all the "easy" homologs in the Later Set by using the 30% sequence identity clusters of all protein sequences provided by the PDB. For a complex in the Later Set to be considered further, at least one protein chain of this complex must not be found within the same 30% sequence identity cluster as any chain of any structure in the Earlier Set. This selection procedure left 8544 assembly structures. To further remove the redundancy among them, we applied the same sequence identity criterion and arbitrarily selected 3353 structures that have at least one chain not found in the same 30% cluster as any chain in any other complex structure within these 8544 structures. A maximum length of 1480 residues was also applied during the selection to the total size of the oligomer (i.e., the residue count of all individual chains) to prevent GPU memory overflow, a limitation imposed by the memory capability of the GPUs available for our tests.

The sequences and coordinates of the 3353 assemblies were subsequently examined. We further require that, in a good target, each protein chain shares a protein-protein interface of at least 20 residues with another chain of the complex. This requirement removed many complexes with short peptides and left 1,884 assembly structures, of which 47 entries with nonstandard amino acids in their PDB SEQRES records were removed, as we used the sequences given by SEQRES as the input sequences to model prediction. The remaining 1838 assemblies consist of 1275 dimers and 563 higher-order oligomers. One entry in the oligomers failed to generate the input during our evaluation of AF-Multimer, and we removed it from further evaluation. These two final sets are called Dimer1275 and Oligomer562, respectively. Because the mapping of chains between model and experimental structure is an open issue for automated assessment of high-order oligomers, primarily due to the combinatorial growth of identical or homologous copies, we mainly focused on the dimer set in this study. However, we provide an evaluation of the physicality of the predicted Oligomer562 models.

In addition, AF-Multimer uses different template identification protocols between the multimer and monomer predictions. The multimer data pipeline uses a protocol that searches essentially all PDB structures up to an optional cutoff date (Apr. 30, 2018, in our benchmark scenario), but the monomer protocol searches only a subset of PDB structures (PDB70, representing single chains at 70% sequence identity cutoff). All targets of Dimer1275 have identified four templates in the multimer protocol, but 82 targets have none or fewer than four templates identified in the monomer protocol that AF2Complex relies on for input generation. To alleviate this unfairness, we removed these 82 dimers, leading to the final Dimer1193 set for the benchmark tests. The union of Dimer1193 and Oligomer562 are named as Oligomer1755.

**E. coli sets**. The sequence of E. coli strain K12 was downloaded from UniProt[58] (Proteome ID UP000000625). We used a positive and negative protein-protein interaction set curated previously[27,32]. We first filtered out pairs whose total size is longer than 1,480 residues, which is a limit imposed by the 16 GB GPU memory per node on the Summit supercomputer. Filtering led to 701 pairs from the positive set. Because the original negative set is too large to run all, we randomly selected 6,849 pairs from them, which yielded roughly a 1:10 ratio between the positive and negative set.

**E. coli CcmI system modeling**. Three rounds of modeling were performed for this system composed of eight proteins, CcmA, B, C, D, E, F, G, and H. In the first round, based on the literature, we tested 22 combinations of these proteins. Among them, top models with high confidence interface scores and literature corroborations were presented for three assemblies, CcmA$_2$B$_2$CDE, CcmEFH, and CcmFGH. All models were generated with the AF2 monomer DL models by applying up to 20 recycles. Application of the AF-Multimer DL models with AF2Complex did not yield better models with higher confident scores and are thus not presented.

In the second round, we tested the interactions between various apocyt c peptides and the CcmFGH complex. A cytochrome c protein, NrfA of E. coli, was arbitrarily chosen for modeling. NrfA contains five CXXC motifs for covalent heme attachment[59]. We cropped peptides spanning the heme-binding motifs, including 8 peptides with one CXXC motif and three with two motifs. The lengths of these apocyt c peptides range from 8 to 52 residues, mostly around 18 AAs. During modeling, an apocyt c substrate and CcmFGH were folded simultaneously by the DL models, in contrast to typical docking, whereby a substrate is placed into a putative binding site of a folded protein or complex structure. By simultaneously modeling both the receptor and the acceptor, one might obtain a better complex structure that requires large conformational changes due to interactions. Note that we did not re-generate the MSAs for each peptide substrate. Rather, we cropped out the input features for each peptide from the input features, including the MSAs, of the full NrfA sequence. A domain cropping option implemented in AF2Complex enables this practice, which is convenient and likely more accurate in comparison to regenerating MSAs using partial sequences, especially for short peptides.

In the last round, we modeled heme b molecules in their putative binding sites of CcmC or CcmF of top models. Because AF2 does not currently support the incorporation of cofactors and other prosthetic groups into structural models, we ran the Rosetta relax application[60] using the '-in:auto_setup_metals' option to model heme-bound systems starting from the AF2C models. The 'molfile_to_params.py' script was used to generate the required parameters for heme b. In some cases, the positions of histidine residues were adjusted to place them in proper position for axial coordination prior to Rosetta refinement.

**Performance evaluation**. Standard metrics were applied to the benchmark tests on the CP17 and E. coli sets, both consisting of a true positive and negative set. The predictions were labeled using the pre-defined classification and the numbers of true positives, false positives, true negatives, and false negatives were then designated as TP, FP, TN, and FN, respectively. Performance measures are defined as follows,

$$True\,Positive\,Rate = Recall = \frac{TP}{TP+FN}$$
$$False\,Positive\,Rate = \frac{FP}{TN+FP}$$
$$Precision = \frac{TP}{TP+FP}$$
$$Matthews\,Correlation\,Coefficient = \frac{TP \times TN - FP \times FN}{\sqrt{(TP+FN)(TP+FP)(TN+FP)(TN+FN)}}$$

(4)

We also employed the normalized AUC$_{0.1}$, which is the area under the ROC curve up to an FPR of 0.1, divided by 0.1. ROC curves were plotted using ROCR[61].

**Interface clash indicator**. To characterize the clashes observed in the predicted computational models, we defined an interface clash indicator $\chi \equiv N_{iRes}/N_{iCnt}$, where $N_{iRes}$ is the number of interface residues, and $N_{iCnt}$ is the number of interface residue-residue contacts. An interface residue-residue contact is defined if any heavy-atom of a residue of one chain is within 4.5 Å of any heavy-atom of a residue from another chain. The value of $\chi$ ranges from 0 to 2 with values close to zero indicating a significant number of clashes. For experimental structures, $\chi$ is typically between 0.6 and 1.2. The metric can be directly applied to un-relaxed models to detect severe clashes. In practice, this evaluation can help to save computing time on unphysical models that are unlikely to be fixed by the relaxation protocol of AF2.

**Statistical tests**. To test the hypothesis that the model quality of one model is better than another one in terms of a scoring metric, e.g., the DockQ score, we used the Wilcoxon signed-rank test, a non-parametric statistical test, because the score distributions do not follow normal distributions. The tests were paired, as every method makes a prediction on the same set of targets. All tests are one-tailed.

**Computational costs**. The development tests and predictions on CASP14 assembly targets were carried out locally using about 10 workstations each with four Nvidia RTX6000 GPUs, where each GPU has 24 GB memory. The benchmark tests on CP17 and E. coli sets were performed on the Summit supercomputer at Oak Ridge National Laboratory. A Singularity container was built to run AF2 on Summit[62,63]. An AF2Complex run of ~7000 pairs of proteins using 923 nodes required about 2 hours in wall clock time. Each node has 6 Nvidia 16 GB V100 GPUs. For an individual target of fewer than 1000 residues, models may be obtained within 20 min for each deep learning model using "super" mode, which is a preset of configurations used with AF2Complex for this study.

**Reporting summary**. Further information on research design is available in the Nature Research Reporting Summary linked to this article.

## Data availability

Benchmark data set CP17 were taken from Ref.22. The positive and control test sets of *E. Coli* were derived from Ref.27,32. All benchmark sets, CP17, Dimer1193, Oligomer562, and the full *E. coli* proteome (UniProt Proteome ID UP000000625), including input features to AF2Complex, and the top computational models of *E. coli* Ccm system I generated in this study are available at Zenodo64 (https://doi.org/10.5281/zenodo.6084186).

## Code availability

AF2Complex is open source and available at Github (https://github.com/FreshAirTonight/af2complex) and Zenodo65 (https://doi.org/10.5281/zenodo.6339741).

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

## Acknowledgements

We thank Ada Sedova for coordinating the deployment of AlphaFold2 on Summit at Oak Ridge and critical reading of the manuscript, Ryan Prout, Subil Abraham, Wael Elwasif, N. Quentin Haas for building a Singularity container, and Mark Coletti for providing Dask scripts for running AF2. We thank Jessica Forness for proofreading the manuscript. This work was supported in part by the DOE Office of Science, Office of Biological and Environmental Research (DOE DE-SC0021303, J.S. and J.P.) and the Division of General Medical Sciences of the National Institute Health (NIH R35GM118039, J.S.). The research used resources supported in part by the Director's Discretion Project at the Oak Ridge Leadership Computing Facility, and the Advanced Scientific Computing Research (ASCR) Leadership Computing Challenge (ALCC) program (J.S, J.P, and M.G.). We also acknowledge the computing resources provided by the Partnership for an Advanced Computing Environment (PACE) at the Georgia Institute of Technology.

## Author contributions

M.G. and J.S. designed the research, M.G. and D.A. wrote the source code, M.G. performed research and analyzed the data, M.G. and J.P. analyzed the models of the *E. coli* Ccm system I, M.G. prepared the first draft of the manuscript, M.G., J.P., and J.S. revised the manuscript, and all authors proofread the manuscript.

## Competing interests

The authors declare no competing interests.
