## [Peer Review File · Nature Communications]

Reviewers' Comments:

Reviewer #1:

Remarks to the Author:

Modeling of protein-protein interactions is a highly significant unsolved problem. With the recent mind-boggling advances in accurate modeling of single domain proteins, and to some extent multidomain proteins, this manuscript very timely addresses the next step. Indeed, scientific community is not satisfied with how AF currently models protein-protein interactions, noticing significant imperfections, and even clashes, in such models. The improvements suggested by a new method presented in this manuscript are very encouraging, significant, and immediately useful. This new approach will be welcomed by structural biologists.

A strength of the method is in by-passing paired multiple sequence alignment. Such alignments are frequently difficult to pair in a meaningful way, and it is not clear what sequences should be included in them. The method also estimates the likelihood of the complex formation, which is a really important task. It is nearly always possible to dock several proteins onto each other in some meaningful ways, but the first question biologists want answered is whether the proteins form a biologically functional complex.

The results on a small CASP14 benchmark are very encouraging, and so are larger benchmarks, carefully designed by the authors to find a meaningful compromise between rigor and time invested in computation. The performance achieved indeed seems rather good compared to other similar approaches.

The authors do not mention simple geometric properties of interfaces they model. I know that structural biologists are quite sensitive to seeing clashes in models. Are the models produced clash-free? If not, what would be the expected number of clashes (or other geometric problems) at interfaces?

It is commendable that the authors immediately release the source code to maximize the usefulness of this publication. I have not tried to compile and install the version, but I know from my collaborations with experimental biologists that they would like to use this software in their labs, and with minimal help from outside. Therefore, to ensure wider reach of this publication, I very much encourage the authors to take some extra steps and make compilation and installation of their new software more straightforward, so that interested experimental biologists, with some help from their more computationally inclined students, can use this new tool with relative ease. Most Universities have either their own supercomputers or access to more centralized GPU-intensive resources, thus I expect that there would be a wide community of users, provided that the software is well documented and easy to use.

Reviewer #2:

Remarks to the Author:

Remarks to the authors:

I will focus on the examples of cytochrome c biogenesis complexes (CcmABCDE and CcmFH) since this is my specialty.

However, I do have a few comments on the manuscript concerning the writing:

Seems to me, there are lots of acronyms and specialized language (to the theoretical structure crowd). If Nat Comm is intended as a journal for the general science audience, it would be good to minimize the specialty language if possible. A couple examples but by no means the only:

Line 104... Among the input features, the most critical are the MSAs, which are obtained by extending each monomeric alignment sequence to the full complex length with gap paddings. Correspondingly, to mark separate peptide chains we sequentially increase the residue index feature of the second or later monomer(s) by a large number. The structure templates of monomer sequences are also re-indexed accordingly....

Line 202... The normalized area under the curve (AUC) of this plot, AUC_{0.1}, is 0.72 and 0.69 for S and piTM, versus 0.49 and 0.10 for pTM and pLDDT, respectively. For reference, random guessing yields an AUC_{0.1} of 0.05. The AUC for S is 0.90. Similarly, the maximum of the Matthews correlation coefficients is 0.64 for both S and piTM, compared to 0.47 for the pTM-score and 0.10 for pLDDT, respectively.

Comments on Fig 6 and Ccm structures (System I cyt c biogenesis) as examples:

General comments: The CcmABCDE and CcmEFH complexes are involved in both the trafficking of heme and attachment of heme vinyls to acceptors (CcmE accepts heme from CcmCD; and apocytc (CXXCH) accepts heme from CcmFH, the heme comes from holoCcmE). Thus, where heme binds, its orientation, its access to the complexes, how it is oxidized or reduced, and how the heme-attached acceptor is released from the complexes are major questions. I am not sure that the predicted structures (without heme), as presented, help with these questions, but I keep an open mind on this. A second point concerns the different conformational states that each complex (CcmABCD and CcmFH) has to go through during its cycle. I realize that it is very recent, but a manuscript by Mendez et al in Nat Chem Bio on two cryo-EM structures of the System II CcsBA is instructive: CcsBA transports heme and attaches it to apocytc. Major conformational changes occurred depending on heme. So the lack of substrate (heme) incorporation into structures and the clear likelihood of various conformations may confound the predictions. On the other hand, there is value in the predictions, but the limitations and needs for experimental studies remain critical. I know the authors realize this, but I provide my insights on their Ccm structures here.

Specific comments

CcmABCDE:

1) Looks to me like the CcmE single TM interacts with CcmB, yet it is known from experimental work that the CcmE TM can be exchanged for other TMs and that one can even remove the TM (called CcmE*) and CcmE* still receives heme from CcmABCD or CcmCD. CcmE* is released from the complex upon ATP hydrolysis by CcmA. Please comment on the TM:B interaction.

2) CcmCD forms a tight complex with CcmE when heme is present, and likely a weaker one when heme is not present. Fig 6A(left) shows a complex of CcmABCD with E that has some interesting contacts. These may be useful in the future for genetic experiments to define contact points. However, Fig6A (right) shows that the contacts other than the CcmE TM with CcmB are gone. Given the points about CcmE TM in point 1 above, how is it rationalized that the CcmABCDE complex on the right even forms? What is the value of the Fig6A (right) complex if it is likely incorrect? (That is, is it the same as the left one except CcmE now only has the B interaction with TM).

3) In the Fig6A inset (left) the authors have displayed some important side chains of D (Y17), C (H60&H184) and E (H130 and Y134). These have each been studied with genetic, biochemical, and spectroscopic analyses in published manuscripts. Thus it is likely that they are involved at the active site in heme interaction (eg C H60&H184 form ligands to the heme). So it is interesting that they are near each other in the predicted structure (left). On the other hand, heme is missing in the structures, and critical gaps in the cycle of activity performed by CcmABCDE remain unanswered, do the structures help with any of these questions/answers:

i) CcmE interaction with CcmCD induces access to heme (how does heme access the site, from below, outer leaflet?)

ii) Heme binds in a stable CcmCDE complex, bound by H60 & H184 of CcmC, but also by the WWD domain= heme handling domain the authors refer to. The WWD domain is a conserved domain that is adjacent to the vinyls of heme. Does the structure offer clues on this site?

iii) H130 of CcmE forms a covalent adduct to the vinyl 2 group of heme (where are the vinyls of heme located for this attachment?)

iv) Somehow, ATP hydrolysis by CcmA confers a conformational change to CcmB, likely conferred to CcmC and/D that releases the attached heme. (What structure(s) represent these important intermediates? In particular, exactly what CcmB:C, and/or B:D or C:heme contacts are weakened to release holoCcmE? The authors state on line 271: Interestingly, the absence of CcmE in a separate CcmA2B2CD model led to conformational changes within the complex (Fig. 6A center

panel). Compared to the CcmA2B2CDE model, movement relative to CcmC is evident in CcmA2B2 but not in CcmD, which appears tightly coupled with CcmC. It is hard to see from the figure on whether they have something here that addresses release conformational changes or heme access states. Line 280: We also modeled the CcmA2BCDE complex and found that the configuration of CcmC is similar to that in the CcmA2B2CD complex in that it does not occupy the opening left by a missing CcmB as previously conjectured. Not sure what this means.

Concerning the CcmFH structures in Fig 6B and C:

4) Fig6B inset: It is well known that the CcmF H173 and H303 are ligands to the heme coming from periplasmic holoCcmE (which is attached to H130CcmE). So again it is encouraging that well-studied side chains are close to each other in the CcmEFG complex (I'm not sure where the Y17 comes from?) The Nat Chem Bio paper by Brausemann et al (ref 47) has a crystal structure of CcmF alone (from a thermophile). In that paper and in a N&V commentary, it is proposed that heme in holoCcmE surprisingly enters CcmF from the outer leaflet, not from the soluble periplasm above. In fact this was one of the major conclusions, so can the authors comment on whether their theoretical structures would also suggest this (ie a large pore the size of heme that faces the outer leaflet)?

5) Since the predicted structures here, nor the crystal structure of CcmF (ref 47) do not have periplasmic heme, I am wondering whether a cavity the size of heme is present in these structures, eg next to H173 & 303.

6) Fig 6C on CcmFGH: The thiol reduction motifs (CXXC) of CcmG and H are shown near the CcmF H173 and 303. The authors suggest that they can reduce the apocytochrome c acceptor (CXXCH) prior to attachment. These are interesting theoretical structures, I have no concerns about their guesses on mechanisms here. I am curious if there is any clue as to whether the CXXCH acceptor can fit here, as well as the heme? They mention a "tunnel" but is there room for heme also?

7) CcmF has the WWD domain, is it near a putative heme site?

8) Line 287: The distance between His130 of CcmE and His303 of CcmF is ~ 6 Å (Fig. 6B inset). These conserved histidines are known to coordinate the heme-bound Fe cation. His130 of CcmE is not a heme ligand but is covalently attached to vinyl 2.

9) Line 317: In this case, both proteins instead interact with a third protein, CcmE. The function of the CcmH C-terminal domain remains unclear, but likely involves interactions with an apocytochrome protein. By "both proteins, do they mean CcmF and H or the two CcmH subdomains? I thought that CcmE only docks into CcmF, not requiring CcmH?

Reviewer #3:

Remarks to the Author:

This manuscript studies one way of applying AlphaFold2 to predict protein complex structure and protein-protein physical interactions. The overall idea is to concatenate two proteins as a single one and then apply AlphaFold2 to do prediction. The authors show that such a simple method produce better results than the other ways of applying AlphaFold2 to complex structure prediction.

Overall, the paper is well written, the analysis of the Ecoli proteome and the later analysis of the structural models of the cytochrome c biogenesis system are interesting. Below are some major concerns:

1) But they do not compare their methods with AlphaFold-Multimer.

The benchmark sets they used for testing building 3D models are still quite small. They use CASP14 and the 17 targets used in the AlphaFold-Multimer paper. They probably should use a much larger benchmark set.

2) There is no analysis of the impact of the effective depth and the coverage of MSAs on the quality of the predicted 3D docking models. The scores they proposed to classify interacting pairs

are useful, but they probably should compare with other PPI classifiers and Alphafold-Multimer to show how effective their proposed scores are.

3) Finally, the authors failed to cite a few relevant papers.

3.1) A few years ago Xu group has shown that a deep learning model trained by individual protein chains can be used to predict complex contact. See <https://www.biorxiv.org/node/73722.full> and <https://academic.oup.com/nar/article/46/W1/W432/5001161>. That is, it is not very surprising AlphaFold2 can be used to predict complex structure even trained by individual protein chains.

3.2) Xu group has shown that for many hard test proteins deep learning may predict their tertiary structures well even without using co-evolution information. See <https://www.nature.com/articles/s42256-021-00348-5>. Xu group showed that with only sequence profile (but not coevolution information), deep learning may predict correct folds for more than 50% of the CASP13 FM targets. The result in Xu's paper explains why for some complexes AlphaFold2 still works without paired MSAs. When the two MSAs (of two constituent proteins) are not paired, there is no coevolution information between two proteins, but their sequence profiles still exist.

Point-by-point Response to Reviewers' Comments

We thank all of the reviewers for providing comments that helped us to improve our paper. Below please find a point-by-point response to the reviewers' comments. The response to the j th comment by the i th reviewer is named **R_{i,j}**. For each comment, statements made by a reviewer are quoted first, followed by our reply. In the main manuscript, our revisions are highlighted in blue for your convenience. A new Supplementary Information document has been added for this revision and is referred to below as SI.

Reviewer 1

R1.1 "The authors to not mention simple geometric properties of interfaces they model. I know that structural biologists are quite sensitive to seeing clashes in models. Are the models produced clash-free? If not, what would be the expected number of clashes (or other geometric problems) at interfaces?"

We are grateful for this reviewer's strong encouragement. With respect to this question, we agree with the reviewer's legitimate concern. Indeed, we observed that many clashes at the interfaces of models are generated by AF-Multimer (AlphaFold version 2.1.1) using its default settings. These phenomena are demonstrated in a new analysis on a set of 563 recently released oligomers (≥ 3 chains), on which about 20% of AF-Multimer's top model have severe clashes (see SI for an example and statistics provided in Fig. S3). The issue is somewhat alleviated by 14%, if we use the same AF-Multimer deep learning model, but our approach uses unpaired multiple sequence alignments (MSAs) instead of paired MSAs. Furthermore, with the original AF2 monomer deep learning models and unpaired MSAs, although it could miss some good predictions only seen with the new multimer models, the predicted models appear to be much more physical and more closely resemble experimental structures. There could still be minor clashes as observed in AF2's models of single protein chains, but these can be removed largely by the molecular dynamics minimization protocol of AF2. In short, our new analysis reveals a shortcoming of AF-Multimer (AF ver. 2.1.1) and the significant benefits of using unpaired MSAs with AF2Complex with both AF2 monomer (AF ver. 2.0.1) and multimer (AF ver. 2.1.1) deep learning models. To aid the analysis, we introduced a convenient metric, the interface clash indicator χ , to estimate the severity of clashes in a predicted complex structural model.

Reviewer 2

R2.1 "General comments: The CcmABCDE and CcmEFH complexes are involved in both the trafficking of heme and attachment of heme vinyls to acceptors (CcmE accepts heme from CcmCD; and apocyt c (CXXCH) accepts heme from CcmFH, the heme comes from holoCcmE). Thus, where heme binds, its orientation, its access to the complexes, how it is oxidized or reduced, and how the heme-attached acceptor is released from the complexes are major questions. I am not sure that the predicted structures (without heme), as presented, help with these questions, but I keep an open mind on this. A second point concerns the different conformational states that each complex (CcmABCD and CcmFH) has to go through during its cycle. I realize that it is very recent, but a manuscript by Mendez et al in Nat Chem Bio on two cryo-EM structures of the System II CcsBA is instructive: CcsBA transports heme and attaches it to apocyt c. Major conformational changes occurred depending on heme. So the lack of substrate (heme) incorporation into structures and the clear likelihood of various conformations may confound the predictions. On the other hand, there is value in the predictions, but the

limitations and needs for experimental studies remain critical. I know the authors realize this, but I provide my insights on their Ccm structures here.”

We agree with the reviewer that the lack of heme in our previous version of the manuscript was not ideal. In this revision, we have performed extensive modeling of heme-bound complexes to generate models that are as complete and realistic as possible. Because AF2 is not currently able to incorporate ligands, cofactors, or other prosthetic groups, we used a conventional modeling approach to perform these steps on the AF2Complex models. In addition, we have modeled apo-cytochrome *c* peptides containing CXXCH motifs bound to the CcmFGH proteins using our deep learning method AF2Complex, rather than standard docking approaches. These modeling efforts, together with the originally presented models, provide insights into Ccm system I. We further address the specific questions raised by the reviewer individually below. We also thank the reviewer for bringing the CcsBA work to our attention. The heme-bound CcsBA structures indeed lend key insights into a similar heme-binding pocket in one of our complex models.

R2.2 “1) Looks to me like the CcmE single TM interacts with CcmB, yet it is known from experimental work that the CcmE TM can be exchanged for other TMs and that one can even remove the TM (called CcmE*) and CcmE* still receives heme from CcmABCD or CcmCD. CcmE* is released from the complex upon ATP hydrolysis by CcmA. Please comment on the TM:B interaction.”

We present two top complex models of CcmA₂B₂CDE. In one of them, CcmE has many interactions with both CcmC and one CcmB, as shown in the revised Fig. 6c. The interactions between CcmE and CcmC are in the His130-containing loop of CcmE and in many residues of CcmC including Trp114 and Trp119 from the so-called WWD domain (which is mostly a single loop in our model). These interactions between CcmC and E are transient as they were almost entirely absent in the other CcmA₂B₂CDE model in which CcmE is largely detached from the assembled complex. However, the most important contribution to the tight binding between CcmC and E is likely due to the covalent attachment of the heme in the CcmC to CcmE. The energy released by ATP hydrolysis may be critical to the release of holo-CcmE. Hypothetical conformational changes due to ATP hydrolysis are displayed in the two models we presented.

On Page 8, we added,

“Extensive contacts are present in the model with CcmC tightly bound with CcmE. The interactions involve Trp114 and Trp119 of a tryptophan-rich loop (WWD domain⁴⁸), sitting at one edge of the binding pocket of the heme (Fig. 6c).”

R2.3 “2) CcmCD forms a tight complex with CcmE when heme is present, and likely a weaker one when heme is not present. Fig 6A(left) shows a complex of CcmABCD with E that has some interesting contacts. These may be useful in the future for genetic experiments to define contact points. However, Fig6A (right) shows that the contacts other than the CcmE TM with CcmB are gone. Given the points about CcmE TM in point 1 above, how is it rationalized that the CcmABCDE complex on the right even forms? What is the value of the Fig6A (right) complex if it is likely incorrect? (That is, is it the same as the left one except CcmE now only has the B interaction with TM).”

As the reviewer mentioned above in R2.1 and below in R2.5, CcmE is released from the CcmABCD complex upon ATP hydrolysis by CcmA. Therefore, the interactions between CcmE and the complex are transient. The model in question represents a conformation of

CcmABCD (almost) without interactions with CcmE, though we cannot tell exactly whether this model represents a state before or after heme loading. Nevertheless, it is a hypothetical model corroborating known experimental evidence and could represent the conformation after CcmE retrieved the heme from CcmC. We have revised the relevant text to better clarify this point. Moreover, these two models represent an example of multiple (likely biologically relevant) conformations of a target complex obtained by applying AF deep learning models. This finding is of general interest to many research groups with implications beyond the specific molecular system we studied.

R2.4 “3) In the Fig6A inset (left) the authors have displayed some important side chains of D (Y17), C (H60&H184) and E (H130 and Y134). These have each been studied with genetic, biochemical, and spectroscopic analyses in published manuscripts. Thus it is likely that they are involved at the active site in heme interaction (eg C H60&H184 form ligands to the heme). So it is interesting that they are near each other in the predicted structure (left). On the other hand, heme is missing in the structures, and critical gaps in the cycle of activity performed by CcmABCDE remain unanswered, do the structures help with any of these questions/answers:)

i) CcmE interaction with CcmCD induces access to heme (how does heme access the site, from below, outer leaflet?). ii) Heme binds in a stable CcmCDE complex, bound by H60 & H184 of CcmC, but also by the WWD domain= heme handling domain the authors refer to. The WWD domain is a conserved domain that is adjacent to the vinyls of heme. Does the structure offer clues on this site? iii) H130 of CcmE forms a covalent adduct to the vinyl 2 group of heme (where are the vinyls of heme located for this attachment?)”

We thank the reviewer for these insightful questions. We have modeled a heme molecule into the proposed binding pocket (Fig. 6d & e). To the best of our knowledge, the heme-bound model is consistent with known biochemical data, e.g., the heme Fe cation is coordinated by His60 and His184. To answer question (i), we do see a pore between helices 2, 5, and 6 that allows heme access from the outer leaflet, as shown in Fig. 6e, but we do not see a channel within CcmC that would allow heme access from the cytoplasmic side like CcsBA does. However, we note that after the heme is attached to CcmE, the heme cannot be released through the same pore according to the model, because a loop (residues 180 to 195 of CcmC) would block heme re-entrance to the outer leaflet. CcmE must carry the heme into the soluble periplasm, at least initially after the retrieval. For (ii), in our model, the WWD domain is largely a loop lining part of the binding-pocket to the heme. Two Trp residues, Trp119 and Trp123, make contact with the heme in the docking model, whereas Trp122 and Trp125 make hydrophobic contacts with CcmB. For (iii), we have marked the location of the 2-vinyl (renamed 3-vinyl to follow the IUPAC standard) in the new Fig. 6d. His130 of CcmE is about 4 Å away from the 3-vinyl group in the heme-bound model.

On Page 8, we added,

“A pore is visible between transmembrane helices 2 and 5 of CcmC and exposes a heme molecule bound to CcmC (Fig. 6d). The pore may allow the access of a heme from the outer leaflet, and no channel for potential heme trafficking is present within CcmC as previously speculated⁴⁸. The role of His60^{CcmC} and His184^{CcmC}, predicted heme iron-coordinating residues, is confirmed as well (Fig. 6e). His130^{CcmE} is only ~4 Å away from the 3-vinyl of heme (IUPAC numbering standard), which is the proposed site of covalent attachment to His130^{CcmE} to complete the heme delivery to CcmF.”

R2.5 “iv) Somehow, ATP hydrolysis by CcmA confers a conformational change to CcmB, likely

conferred to CcmC and/D that releases the attached heme. (What structure(s) represent these important intermediates? In particular, exactly what CcmB:C, and/or B:D or C:heme contacts are weakened to release holoCcmE? The authors state on line 271: Interestingly, the absence of CcmE in a separate CcmA2B2CD model led to conformational changes within the complex (Fig. 6A center panel). Compared to the CcmA2B2CDE model, movement relative to CcmC is evident in CcmA2B2 but not in CcmD, which appears tightly coupled with CcmC. It is hard to see from the figure on whether they have something here that addresses release conformational changes or heme access states. Line 280: We also modeled the CcmA2BCDE complex and found that the configuration of CcmC is similar to that in the CcmA2B2CD complex in that it does not occupy the opening left by a missing CcmB as previously conjectured. Not sure what this means.”

This question is similar to R2.3, which we have partially addressed. The model shown in the left and right panels of Fig. 6b (originally 6a) corresponds to the model of CcmA₂B₂CD with and without the interactions to CcmE. The conformational changes are displayed in the center panel where we show the global conformational changes. We cannot ascertain that the model on the right corresponds to the detached holo-CcmE, since we do not have a heme in this model. We merely present it as a hypothetical model that might provide useful information for subsequent, detailed analyses by Ccm experts. The “Line 280” comment above addresses the question of whether CcmC interacts with CcmA2B2 or CcmA2B in the same way. The answer is yes according to our modeling. However, we now realize that it had caused confusion and have now removed this sentence.

R2.6 “Concerning the CcmFH structures in Fig 6B and C: 4) Fig6B inset: It is well known that the CcmF H173 and H303 are ligands to the heme coming from periplasmic holoCcmE (which is attached to H130CcmE). So again it is encouraging that well-studied side chains are close to each other in the CcmEFG complex (I’m not sure where the Y17 comes from?) The Nat Chem Bio paper by Brausemann et al (ref 47) has a crystal structure of CcmF alone (from a thermophile). In that paper and in a N&V commentary, it is proposed that heme in holoCcmE surprisingly enters CcmF from the outer leaflet, not from the soluble periplasm above. In fact this was one of the major conclusions, so can the authors comment on whether their theoretical structures would also suggest this (i.e. a large pore the size of heme that faces the outer leaflet)?”

In this revision, we have modeled heme binding in the predicted CcmEFH complex and show that His173 and His303 from CcmF can indeed readily coordinate a heme molecule (P-heme), as can His261 and His461, which coordinate an accessory heme cofactor (TM-heme) as expected (see newly added Fig. 7b). It has been proposed that P-heme is eventually attached to an incoming apo-cytochrome c acceptor protein, whereas TM-heme remains bound to CcmF and helps to “eject” the donor P-heme after its attachment. In our model, the pocket for P-heme is surrounded by transmembrane helices, and we do not see a large pore opening to the outer leaflet as observed in the crystal structure of *Tt*CcmF. We note that a few lipid molecules were co-crystallized next to the P-heme binding pocket in the experimental structure, and the space occupied by the lipids constitutes the pore accessible from the outer leaflet. However, our models do not consider the lipids, and it is possible that side chain packing closes the pore. Also, we note that the crystal structure misses a critical loop that is part of the binding pocket and contains a critical heme-coordinating histidine (equivalent to His303 in CcmF). Additionally, we thank the reviewer for pointing out Y17, which is a typo and should be Y134 of CcmE.

On pages 8 & 9, we added:

“Consistently, a heme-bound model displays the P-heme coordination role of His303^{CcmF} and His173^{CcmF}, and the TM-heme is coordinated by His261^{CcmF} and His461^{CcmF} (Fig. 7b). TM-heme was co-crystallized with CcmF from *Thermus thermophilus* (*TtCcmF*) in a recent published X-ray structure⁴⁹. Without using this structure as a template, our predicted CcmF model has a TM-score of 0.92 compared to the experimental structure. However, a critical loop containing the equivalent histidine of *E. coli* His303^{CcmF} is missing in the *TtCcmF* structure, suggesting the P-heme binding pocket might be flexible. Indeed, only one of the two expected tryptophans (W229 and W236) from the WWD domain is in contact with the P-heme in the heme-bound model. Moreover, CcmF in our model does not expose a pore as observed the *TtCcmF* structure, in which the pore was speculated to enable P-heme access⁴⁹. The structural difference may be attributed to lipid molecules occupying the pore in the crystal structure but absent in our computational model.”

R2.7 “5) Since the predicted structures here, nor the crystal structure of CcmF (ref 47) do not have periplasmic heme, I am wondering whether a cavity the size of heme is present in these structures, eg next to H173 & 303.”

Yes. This point is addressed by the new heme-bound model detailed above in R2.5.

R2.8 “6) Fig 6C on CcmFGH: The thiol reduction motifs (CXXC) of CcmG and H are shown near the CcmF H173 and 303. The authors suggest that they can reduce the apocyt c acceptor (CXXCH) prior to attachment. These are interesting theoretical structures, I have no concerns about their guesses on mechanisms here. I am curious if there is any clue as the whether the CXXCH acceptor can fit here, as well as the heme? They mention a “tunnel” but is there room for heme also?”

We appreciate this question, which prompted us to conduct additional modeling of CcmFGH and 11 apocyt peptide fragments taken from an *E. coli* cytochrome *c* protein (detailed in a new section of Methods). Note that we predict the full model of CcmFGH and the apocyt peptide simultaneously with the AF deep learning models, in contrast to a typical docking modeling where a ligand is fit into an already folded receptor. Remarkably, as shown in new Fig. 8b, all top two models of 11 modeling runs pinpoint the same reaction groove we anticipated. We renamed the “tunnel” as a “groove”, which is more geometrically accurate. In addition, we incorporated a heme into the model to confirm the putative donor heme-binding site (P-heme).

On page 9, we added:

“To investigate further, we modeled CcmFGH together with 11 different apocyt, each containing one or two CXXCH motifs (see Methods). Remarkably, in all top models, the apocyt are invariably located in the predicted reaction groove (Fig. 8b). Physical contact between Cys46^{CcmH} and an apocyt cysteine is present in some models (Fig. 8c). We further modeled a heme in the expected P-heme site of CcmF using one model. The heme-bound model is largely as expected, but more conformational changes are necessary for the apocyt-heme attachment, as the cysteines of the apocyt are still ~16 Å away from the 8-vinyl group of the P-heme. The speculation echoes large conformational changes upon heme-binding revealed in recently determined structures of CcsBA, a cytochrome *c* synthetase from a different Ccm system⁵⁰.”

R2.9 “7) CcmF has the WWD domain, is it near a putative heme site?”

Yes. The WWD loop surrounds the binding site for the P-heme. One of the tryptophans, W229, makes contact with the P-heme in our heme-bound models (Fig. 7b and Fig. 8c).

R2.10 “8) Line 287: The distance between His130 of CcmE and His303 of CcmF is ~ 6 Å (Fig. 6B inset). These conserved histidines are known to coordinate the heme-bound Fe cation. His130 of CcmE is not a heme ligand but is covalently attached to vinyl 2.”

We thank the reviewer for pointing out this omission. We now show His130^{CcmE} and the vinyl group of the heme in the new Fig. 6d. We also revised the second sentence above as: “His303 is highly conserved and known to coordinate the heme-bound Fe cation.”

R2.11 “9) Line 317: In this case, both proteins instead interact with a third protein, CcmE. The function of the CcmH C-terminal domain remains unclear, but likely involves interactions with an apocyt protein. By “both proteins, do they mean CcmF and H or the two CcmH subdomains? I thought that CcmE only docks into CcmF, not requiring CcmH?”

We apologize for the confusion. By “both proteins” we meant “both CcmH domains”. This mistake has been clarified in the revision. The reviewer is correct that CcmE can interact with CcmF with or without CcmH, and we observed a similar quaternary arrangement between CcmE and F in our computational modeling. Interestingly, the N-term domain of CcmH seems to have multiple binding poses with CcmF, dependent on the presence of CcmE, as we describe in the text.

R2.12 “Seems to me, there are lots of acronyms and specialized language (to the theoretical structure crowd). If Nat Comm is intended as a journal for the general science audience, it would be good to minimize the specialty language if possible. A couple examples but by no means the only:

Line 104... Among the input features, the most critical are the MSAs, which are obtained by extending each monomeric alignment sequence to the full complex length with gap paddings. Correspondingly, to mark separate peptide chains we sequentially increase the residue index feature of the second or later monomer(s) by a large number. The structure templates of monomer sequences are also re-indexed accordingly....

Line 202... The normalized area under the curve (AUC) of this plot, AUC_{0.1}, is 0.72 and 0.69 for S and piTM, versus 0.49 and 0.10 for pTM and pLDDT, respectively. For reference, random guessing yields an AUC_{0.1} of 0.05. The AUC for S is 0.90. Similarly, the maximum of the Matthews correlation coefficients is 0.64 for both S and piTM, compared to 0.47 for the pTM-score and 0.10 for pLDDT, respectively.”

We apologize for the acronyms and technical language. We strive to make our writing as accessible as possible. In this revision, we have made editorial changes to improve the text. For example, we have revised the explanation of modifying the residue index. In simple terms, we just need to add a gap in the residue numbering to signify separate chains. In the paragraph the reviewer referred to above, we have removed two sentences about AUC and Matthews correlation coefficients and left only descriptions about AUC_{0.1}, which is the quantity most relevant to practical applications as the imbalanced sets (much more true negatives than true positives) have practical implications. We also spelled out “S” (as interface-score).

We also added a cartoon illustration of the Ccm system I to help readers unfamiliar with this complex system.

Reviewer 3

R3.1 “1) But they do not compare their methods with AlphaFold-Multimer. The benchmark sets they used for testing building 3D models are still quite small. They use CASP14 and the 17 targets used in the AlphaFold-Multimer paper. They probably should use a much larger benchmark set.”

In this work, one major focus is to predict protein-protein interactions, not just on the quality of predicted complex models given known interacting proteins. As such, in addition to the small-scale benchmark tests the reviewer mentioned, we did conduct a large-scale test on ~7,500 protein pairs from the *E. coli*. Moreover, we consider the development of our approach in parallel to AlphaFold-Multimer, whose source code was released when we were finalizing our manuscript and only one week prior to the first release of our source code. We note that the AF-Multimer work was solely focused on predicting model quality given a set of intermolecular interactions, rather than predict interactions from arbitrary protein pairs as we pursued. Therefore, there is a significant conceptual difference between these two works. Our approach is not necessarily in direct competition to AF-Multimer, but rather a further development that takes advantage of various AF2 deep learning models.

Having said the above, we share reviewer’s concern about benchmark size and the interest in a comparison with AF-Multimer. We must point out that the CP17 benchmark set was taken from an earlier docking study (Reference 21), not from the AF-Multimer preprint as the reviewer stated. The same set was used by both DeepMind and us for a convenient comparison with the docking-centric approaches. It is not straightforward to create an objective benchmark data set, because the training set of AF2 deep learning models have never been released for any AF version. We note that a large benchmark set was created by DeepMind for AF-Multimer evaluation according to its preprint. However, this data set was not publicly available. In this revision, in addition to a comparison to AF-Multimer on the CP17 set, we have carefully curated a large benchmark data set of ~1,800 protein complex structures recently released in the PDB. It composed of ~1,200 dimers and ~600 higher-order oligomers. Due to the length limitation, we have to place the results of our new benchmark test in the SI (and the curation of the benchmark set in Methods). A succinct summary of new benchmark results is provided on page 6 as the follows,

“Moreover, using AF2 monomer DL models, AF2Complex performs quite close to AF-Multimer³⁵ on the CP17 set (Fig. 3c), with a mean DockQ score of 0.62 versus 0.65. The mean score of AF2Complex increases to 0.74 using AF-Multimer DL models (Fig. 3d), thanks to unpaired MSAs, increased recycles and the interface-score metric (Supplementary Table 2). By combining the overall top model from AF2Complex runs using either monomer and multimer DL models of AF, we obtain acceptable or better models for all 17 targets.

To address the small target size of CP17, a large-scale benchmark study was further carried out on 1,192 dimers and 593 oligomers (see Methods and SI). On 440 heterodimers obtained using the same DL models, AF2Complex performs significantly better than AF-Multimer, albeit at smaller advantage (median/mean 0.69/0.56 versus 0.65/0.55, p -value = 5×10^{-3} , see SI and Fig. S2), whereas their performance on the homodimer set is comparable (Fig. S2). AF2Complex further improves (median/mean 0.70/0.57 on the heterodimers, p -value = 6×10^{-4}) if both monomer and multimer DL models are assessed. It must be pointed that AF-Multimer (version 2.1.1) may yield unphysical models with severe clashes for a complex, especially for large oligomeric targets including homo-oligomers. In contrast, AF2Complex

mitigates this issue with unpaired MSAs using the same multimer DL models, and deals much more effectively with unpaired MSAs and the original monomer DL models of AF2 (SI and Fig. S3).”

R3.2 “2) There is no analysis of the impact of the effective depth and the coverage of MSAs on the quality of the predicted 3D docking models. The scores they proposed to classify interacting pairs are useful, but they probably should compare with other PPI classifiers and AlphaFold-Multimer to show how effective their proposed scores are.”

The impact of MSAs on the quality of predicted models by AlphaFold2 have been thoroughly investigated in the published Nature work by DeepMind (e.g., Supplementary Figure 11 of Ref. 1). It appears that more than 30 sequences in MSAs already delivers a good chance of success. Since our approach employs the very same neural network models, we do not expect to see a much different dependency on MSAs from what was concluded. The exact impact is a topic more suitable in a future study for a specialized journal.

We appreciate the reviewer’s comments with respect to the new metrics that we introduced for both evaluating model qualities and for the predicting protein-protein interactions. To the best of our knowledge, there is no good, existing protein-protein interaction classifiers that can be directly applied to a structural model derived from a deep learning model. We did make comparison to two metrics in pLDDT and pTM provided with the AF models and show the advantage of the new metrics that we introduce (Fig. 4). Overall, the new metrics we proposed are more useful for predicting potential interactions than high ranked structural models of the same target. We did plan to evaluate their performance with the new AF-Multimer models and the metric (iptm+ptm) they proposed. However, AF-multimer has a serious limitation in that it may generate unphysical models with significant clashes within the protein-protein interfaces (see R1.1), which makes it less suitable for predicting protein interactions than the original AF2 models. For now, we think that using the AF2 monomer deep learning models for predicting protein interactions is the best option. And one may use the multimer deep learning models to predict complex models given known interactions. We shall revisit this question in future when better, more stable AF-Multimer neural network models are available.

R3.3 “3) Finally, the authors failed to cite a few relevant papers.

3.1) A few years ago Xu group has shown that a deep learning model trained by individual protein chains can be used to predict complex contact. See <https://www.biorxiv.org/node/73722.full> and <https://academic.oup.com/nar/article/46/W1/W432/5001161>. That is, it is not very surprising AlphaFold2 can be used to predict complex structure even trained by individual protein chains.

3.2) Xu group has shown that for many hard test proteins deep learning may predict their tertiary structures well even without using co-evolution information. See <https://www.nature.com/articles/s42256-021-00348-5>. Xu group showed that with only sequence profile (but not coevolution information), deep learning may predict correct folds for more than 50% of the CASP13 FM targets. The result in Xu's paper explains why for some complexes AlphaFold2 still works without paired MSAs. When the two MSAs (of two constituent proteins) are not paired, there is no coevolution information between two proteins, but their sequence profiles still exist.”

We thank the reviewer for bringing these three references to our attention. The first two of

them are essentially the same work; as such we cited in the revised Introduction. The third reference is now added to the Discussion.

In Reference, we added:

“18 Zhou, T.-m., Wang, S. & Xu, J. Deep learning reveals many more inter-protein residue-residue contacts than direct coupling analysis. *bioRxiv*, 240754 (2017).

51 Xu, J., McPartlon, M. & Li, J. Improved protein structure prediction by deep learning irrespective of co-evolution information. *Nature Machine Intelligence* **3**, 601-609 (2021).”

Reviewers' Comments:

Reviewer #2:

Remarks to the Author:

This is a revised manuscript that I previously reviewed. I analyzed their new changes and additions (blue highlighted), as well as their revised figures on Ccm structures. I also appreciated their responses to reviewers. The authors have responded nicely to all my comments and I believe the Ccm structures will be more useful to the cytochrome c biogenesis field. It will be very interesting to evaluate their structures with experimental structures in the future.

Reviewer #3:

Remarks to the Author:

I am satisfied with the revision.